# Calibrating Radar Wind Profiler Reflectivity Factor using Surface Disdrometer Observations

Christopher R. Williams[1], Joshua Barrio[1], Paul E. Johnston[2,3], Paytsar Muradyan[4], and Scott E. Giangrande[5]

[1]Smead Aerospace Engineering Sciences Department, University of Colorado, Boulder, CO, 80303, USA
[2]Cooperative Institute for Research in Environmental Sciences, University of Colorado, Boulder, CO, 80316, USA
 NOAA Physical Sciences Laboratory, Boulder, CO, 80305
[4]Argonne National Laboratory, Lemont, Illinois, 60439, USA
[5]Environmental and Climate Sciences Department, Brookhaven National Laboratory, Upton, NY, 11793, US

*Correspondence to*: Christopher R. Williams (Christopher.Williams@colorado.edu)

**Abstract.** This study uses surface disdrometer reflectivity factor estimates to calibrate the vertical and off-vertical pointing radar beams produced by an Ultra High Frequency (UHF) band radar wind profiler (RWP) deployed at the US Department of Energy (DOE) Atmospheric Radiation Measurement (ARM) program Southern Great Plains (SGP) Central Facility in northern Oklahoma from April 2011 through July 2019. The methodology consists of five steps. First, the recorded Doppler velocity power spectra are adjusted to account for Nyquist velocity aliasing and coherent integration filtering effects. Second, the spectrum moments are calculated. The third step increases the signal-to-noise ratio (SNR) due to inflated noise power estimates during convective rain events that cause SNR to be biased low. The fourth step determines the RWP calibration constant for one radar beam (called the "reference" beam) by comparing uncalibrated RWP reflectivity factors at 500 m above the ground to 1-min resolution surface disdrometer reflectivity factors. The last step uses the calibrated reference beam reflectivity factor to calibrate the other radar beams during precipitation. There are two key findings. The RWP sensitivity decreased approximately 3-to-4 dB/year as the hardware aged. This drift was slow enough that the reference calibration constant can be estimated over 3-month intervals using episodic rain events. Calibrated moments are available on the DOE ARM data archive and Python processing code is available on public repositories.

## 1 Introduction

Ultra High Frequency (UHF) band (900 – 1290 MHz) radar wind profiler (RWP) technology was developed in the 1980s by the U.S. National Oceanic and Atmospheric Administration (NOAA) Aeronomy Laboratory and Wave Propagation Laboratory to study the horizontal wind motions from near the surface to approximately 5 km above ground level (Ecklund et al., 1988; Angevine et al., 1996, 1998; Carter et al., 1995). When raindrops are not in the radar resolution volume, the radar return power during this "clear-air" condition is due to Bragg scattering from changes in refractive index caused by temperature and humidity gradients (Gage and Balsly, 1978). When raindrops are in the radar resolution volume, the long radar wavelength of

0.25 to 0.33 m implies that Rayleigh scattering dominates the return signal providing vertical structure of precipitation without any signal attenuation (Rogers et al., 1993). Calibration procedures for radars operating at higher frequencies will need to account for attenuation through the precipitation (Williams, 2022).

Radars measure the return signal power as a function of range. For meteorological applications, the signal power needs to be converted to radar reflectivity factor. In general, there are two methods to convert signal power to radar reflectivity factor. The first method directly converts the measured power and range information into radar reflectivity factor. This method requires rigorous characterisation of every radar hardware component using best engineering practices. For radars with steerable antennas, rigorous engineering practices include recording the transmitted power in real-time and performing balloon

mounted sphere calibrations to characterize the antenna beam pattern and beam pointing hardware (Chandrasekar et al., 2015). For radars that are not end-to-end rigorously characterized (e.g., radar wind profilers), the radar reflectivity factor can be estimated indirectly by using the noise relative signal power (i.e., signal-to-noise ratio SNR) and an external reference to determine the radar calibration constant. For vertically pointing radars, the external reference has come from ground-based radars (e.g., Hogan et al., 2000; Williams 2012; Kneifel et al., 2015; Radenz et al., 2018), from near-by surface disdrometer

observations (Gage et al., 2000; Williams et al., 2005; Myagkov et al., 2020), from near-by rain gauges (Hartten et al. 2019), and from satellite radar statistics (Protat et al., 2011; Kollias et al., 2019; Hartten et al., 2019; Protat et al., 2022).

      Since RWPs were originally designed for horizontal wind profile measurements, the NOAA Doppler velocity power spectra processing routines were optimized to estimate mean radial velocity and did not estimate radar reflectivity factor (Merritt, 1995). Even today, real-time processed NOAA RWP datasets do not estimate radar reflectivity factor, but include the

spectrum moments of SNR, mean radial velocity, spectrum width, and noise power (NOAA, 2022). The radar reflectivity factor is estimated from SNR as shown in Gage et al. (1994, 2000) and described in more detail in Tridon et al. (2013) and Hartten et al. (2019). One limitation of RWP signal processing routines is that increased noise power occurs at range gates that have large backscattered signal power. This over-estimated noise power leads to under-estimated SNR, which leads to under-estimated radar reflectivity factor. The elevated noise power in RWPs was discussed in Tridon et al. (2013) and mitigated by

using the measured noise power at far range gates as a new noise power at all range gates. The adjusted SNR is then used to estimate the radar reflectivity factor. The work presented herein builds on the concepts discussed in Tridon et al. (2013), but includes additional SNR biases not discussed in that work. Specifically, this study includes signal power biases due to Nyquist velocity aliasing and coherent integration filtering. Also, this study uses a daily median noise power in the adjusted SNR estimate to account for RWP operating modes that do not have range gates sampling above intense precipitation such that the

noise power is still biased high at the "far" range gates.

      As discussed above, an external reference is needed to determine a radar calibration constant and this study uses surface disdrometer reflectivity factors to calibrate RWP radar reflectivity factors obtained at 500 m. The surface disdrometer was about 100 m from the RWP and the calibration procedure includes shifting the time-series data to account for the 500 m vertical displacement and 100 m horizontal separation between the measurement locations. Depending on the wind speed and

direction, disdrometer time-series data could led or lag the RWP time-series data. An overarching aim of this study is to

standardize the RWP signal processing steps to remove known biases in radar reflectivity factor estimates and provide those codes to the radar community on a public repository.

The radar and disdrometer datasets used in this study are described in Section 2 (Data Sets). Spectrum adjustment methods are discussed in Section 3 (Methods) and include adjustments due to Nyquist velocity aliasing, coherent integration filtering, and increased noise power. Section 3 also includes calibration methods derived from surface disdrometer observations. In Section 4 (Results), the radar calibration constant is shown to vary over an 8-year dataset with decreased sensitivity caused by degrading hardware and sudden increases in sensitivity due to installing new hardware. Conclusions are presented in Section 5 and Appendix A provides additional processing code details.

## 2 Data Sets

This study uses radar observations from a UHF-band radar wind profiler (RWP) operating at 915 MHz and a surface disdrometer located at the US Department of Energy (DOE) Atmospheric Radiation Measurement (ARM) program (Mather and Voyles, 2013) Southern Great Plains (SGP) Central Facility in northern Oklahoma, USA, from 22-March-2011 to 18-August-2019. All datasets used in this study are available online using the ARM Data Discovery Tool (ARM 1998a, 1998b, 1998c, 1998d, 2011).

### 2.1 Radar Wind Profiler

The ARM SGP Central Facility RWP was a Vaisala Meteorological Systems Inc. LAP-3000 wind profiler (Muradyan and Coulter, 2020) and is a commercial version of the NOAA UHF wind profiler developed under an industry-government 1991 Cooperative Research and Development Agreement (CRADA) (Vaisala News, 2002). From 22-March-2011 to 31-March-2014, the RWP operated in a *precipitation mode* observing only in the vertical direction. The precipitation mode sampled the atmosphere with a short- and long-pulse yielding low-sensitivity short-range measurements and high-sensitivity long-range measurements, respectively. On 1-April-2014, a *wind mode* was added to the RWP and consisted of transmitting pulses in three different directions in order to estimate the horizontal wind as a function of height. The RWP collected data in both precipitation and wind modes for 5 years. On 11-March-2019, the wind mode operating parameters changed and on 19-August-2019, the RWP hardware failed and was eventually replaced with a wind profiler produced by a different radar manufacturer. The LAP-3000 RWP can only collect data in one beam direction with one pulse configuration at a time. Thus, during the 2011-to-2014 period, the radar alternated between two vertically pointing precipitation mode radar beams, requiring approximately 5 seconds to collect both beams of data. During the 2014-to-2019 period, the radar sequentially collected data in five unique radar beams (i.e., two precipitation mode beams and three wind mode beams), requiring approximately 25 seconds to complete one observation cycle. Table 1 lists pertinent RWP operating parameters for both modes.

The ARM RWP uses the manufacturer's default processing routines (Muradyan and Coulter, 2020). For each mode,

the RWP transmits a sequence of pulses and performs coherent integrations, Fast Fourier Transforms (FFTs), and spectra averages. Using the precipitation short-pulse mode as an example, the RWP transmits 56 radar pulses (represented by $N_{coh}$) and integrates the in-phase and quadrature voltages (also called $I$ and $Q$ voltages) to produce one in-phase and one quadrature voltage (i.e., $I_{coh}$ and $Q_{coh}$). After collecting 128 (represented by $N_{pts}$) coherently averaged $I_{coh}$ and $Q_{coh}$ voltages, a von Hann window is applied to the time series and a complex FFT is performed to produce a Doppler velocity power spectrum. Another sequence of 7,168 pulses (calculated as $N_{coh}N_{npts}$) are transmitted and processed to produce another Doppler velocity power spectrum. After producing 3 power spectra (represented by $N_{spc}$), the 3 power spectra are averaged and saved to disk. The option of calculating a median spectrum or statistically averaging the 3 spectra (as discussed in Merritt, 1995) in order to remove transient signals (e.g., birds or other flying objects passing through the radar beam) was not implemented. A total of 21,504 pulses $(N_{coh}N_{pts}N_{spc})$ are transmitted per dwell and a 100 $\mu$s inter-pulse period yields a 2.2 s dwell. For each Doppler velocity spectrum, the first three spectrum moments (i.e., signal-to-noise ratio, mean radial velocity, and spectrum width) are estimated using the manufacturer's single peak processing routine with integration limits bounded by the Nyquist velocities $\pm V_{Nyquist}$. The average spectra and the moments are saved to disk.

Between 2011 and 2019, the RWP had two hardware failures. In 2015, the phase shifter module controlling the beam pointing direction failed due to age and overuse. A new phase shifter module was installed. In 2017, the final amplifier in the transmitter module failed and several relays failed in the phase shifter module. The transmitter module was replaced with a used Vaisala unit scavenged from a newer RWP and the relays were replaced. Since calibration constants change with ageing and changing hardware, the RWP dataset is divided into five calibration periods as listed in Table 2.

**Table 1. Pertinent RWP operating parameters (SGP, Central Facility, 22-March-2011 through 18-August-2019).**

| | Precip. Short-Pulse | Precip. Long-Pulse | Wind Mode | | |
|---|---|---|---|---|---|
| Operating Frequency [MHz] | 915 | | | | |
| Operating Wavelength [m] | 0.328 | | | | |
| | | | BeamV | BeamA | BeamB |
| Observation Start Date | 22-March-2011 | 22-March-2011 | 1-April-2014 | | |
| Observation End Date | 18-August-2019 | 18-August-2019 | 10-March-2019 | | |
| Pulse duration [ns] | 417 | 2833 | 708 | 708 | 708 |
| Range Resolution [m] | 62.5 | 425 | 106 | 106 | 106 |
| Distance between Range Gates [m] | 125 then 62.5* | 212.5 | 62.5 | 62.5 | 62.5 |
| Number of Range Gates | 75 then 150* | 75 | 60 | 60 | 60 |
| Range to First Gate [m] | 327 | 327 | 373 | 373 | 373 |
| Range to Last Gate [km] | 9.6 | 16.0 | 4.0 | 4.0 | 4.0 |
| Elevation Angle [degree] | 90 | 90 | 90 | 77 | 77 |
| Azimuth Angle [degree] | 22 | 22 | 22 | 22 | 292 |
| Inter-pulse Period (Tipp) [µs] | 100 | 120 | 41 | 41 | 41 |
| Number of Coherent Integrations ($N_{coh}$) | 56 | 34 | 200 | 200 | 200 |
| Number of points in spectrum ($N_{pts}$) | 128 | 128 | 64 | 64 | 64 |
| Number of Averaged spectra ($N_{spc}$) | 3 | 4 | 12 | 12 | 12 |
| Number of transmitted pulse per dwell^ | 21,504 | 17,408 | 153,600 | 153,600 | 153,600 |
| Nyquist Velocity ($V_{Nyquist}$) [m s⁻¹] | 14.6 | 19.6 | 9.99 | 9.99 | 9.99 |
| Velocity resolution ($v$) [m s⁻¹] | 0.228 | 0.306 | 0.312 | 0.312 | 0.312 |
| Dwell⁺ [s] | 2.2 | 2.1 | 6.3 | 6.3 | 6.3 |

\* Distance between range gates and the number of range gates changed on 4-April-2014

^ Number of transmitted pulses per dwell: $(N_{coh}N_{npts}N_{spc})$

⁺ Dwell is the time needed to transmit all pulses: Dwell $= (T_{ipp}N_{coh}N_{pts}N_{spc})$ [s]

**Table 2. RWP operating periods with consistent hardware**

| Period | Start | End | Hardware Version | Operating Modes |
|--------|-------|-----|------------------|-----------------|
| A | 22-March-2011 | 31-March-2014 | Radar hardware #1 | Precipitation |
| B | 1-April-2014 | 14-July-2015 | Radar hardware #1 | Precipitation and Wind |
| - | 15-July-2015 | 24-Sept-2015 | Hardware failure | No data collected |
| C | 25-Sept-2015 | 10-April-2017 | Radar hardware #2 | Precipitation and Wind |
| - | 11-April-2017 | 5-June-2017 | Hardware failure | No data collected |
| D | 6-June-2017 | 10-March-2019 | Radar hardware #3 | Precipitation and Wind |
| E | 11-March-2019 | 18-August-2019 | Radar hardware #3 | Precipitation |

## 2.2 Surface Disdrometer

A 2-dimensional video disdrometer (VDIS) manufactured by Joanneum Research, in Graz, Austria (Schönhuber et al., 2008), was deployed about 100 m from the RWP at the SGP Central Facility (Wang et al., 2021; ARM, 2011). The 2DVD uses two orthogonal pointing cameras in the horizontal plane to detect raindrops falling through a 10 cm square opening and then estimates the raindrop number concentration with a 1-minute temporal resolution (Tokay et al., 2001, 2013). Radar reflectivity factors assuming Rayleigh scattering were calculated using PyDisdrometer routines (Hardin and Guy, 2014) as used in previous studies using 2DVD observations (Giangrande et al., 2019).

The calibration procedure uses the 1-minute surface disdrometer radar reflectivity factor to estimate a RWP calibration constant for the precipitation short-pulse mode using 1-minute averaged RWP observations at 500 m altitude. The other RWP modes could be calibrated directly with the 1-minute surface disdrometer observations, but to increase the number of samples, the other RWP modes are calibrated using the precipitation short-pulse mode as a reference and using multiple range gates and nearest-in-time observations. The calibration procedure described herein is only valid for RWP modes that collect data while it is raining. If the RWP is adaptive and collects precipitation mode data when it is raining and wind mode data otherwise, then there are not any near-in-time precipitation mode observations nor surface disdrometer observations available to calibrate the wind mode observations. In this situation, the precipitation mode data can be calibrated but the wind mode data cannot be calibrated with the disdrometer observations.

## 3 Methods

The ARM RWP records the average Doppler velocity power spectra and real-time spectrum moments are calculated on the RWP host computer using the RWP manufacturer processing routines. These real-time spectrum moments are labelled "a0" using ARM's file naming protocols (ARM, 2022) and saved on the ARM archive in netCDF format (ARM 1998a, 1998b,

1998c, 1998d). The recorded spectrum moments are not calibrated and do not include a radar reflectivity factor estimate. To illustrate the motivation for reprocessing the recorded spectra and recalculating the spectrum moments, Fig. 1 shows time-height cross-sections of recorded moments including signal-to-noise ratio ($SNR^{a0}$) [dB] (Fig. 1a), mean radial velocity ($V_{mean}^{a0}$) [m s$^{-1}$] (Fig. 1b), and spectrum noise power ($P_{noise}^{a0}$) [dB] (Fig. 1c) for a rain event on 7-June-2018 using the precipitation short-pulse mode. Examination of the $SNR^{a0}$ time-height structure in Fig. 1a suggests convective rain near 11:45 to 12:00 UTC followed by stratiform rain after about 12:15 UTC. There are a couple questionable features in this figure between 11:35 and 12:10 UTC that raise concern about the quality of the real-time spectrum moments. First, the $SNR^{a0}$ contains speckles of low magnitude SNR above the height of about 3 km. Second, the $V_{mean}^{a0}$ has large, unphysical jumps in velocity over several range gates and over several profiles due to Nyquist velocity aliasing. Third, the spectrum noise power $P_{noise}^{a0}$, which is the denominator in estimating SNR, has large and variable magnitudes at nearly all range gates. The first two features are due to the on-line processing codes incorrectly estimating the spectrum moments, and the third feature is due to the broad signal velocity power spectra occupying a large portion of the velocity power spectrum causing the noise level estimate to be contaminated by the signal power.

This section describes the five step RWP calibration procedure. First, the raw Doppler velocity power spectra are adjusted to account for both Nyquist velocity aliasing (see Section 3.1.1) and coherent integration filtering (see Section 3.1.2). Second, the spectrum moments are recalculated (see Section 3.1.3). Third, the recalculated SNR is increased to account for leaking signal power into the noise power to yield an adjusted signal-to-noise ratio (see Section 3.2). Fourth, a calibration constant is determined for the precipitation short-pulse radar beam (defined as the "reference" beam) by comparing radar reflectivity factors with surface disdrometer observations (see Section 3.3). The last step determines relative calibration offsets between the reference beam and the other four radar beams. The calibration constant for each beam is the combination of the reference beam calibration constant and that beam's relative calibration offset (see Section 3.4). To differentiate between the real-time processed moments and the reprocessed moments, the former estimates are labelled "a0" and the latter are labelled "revised".

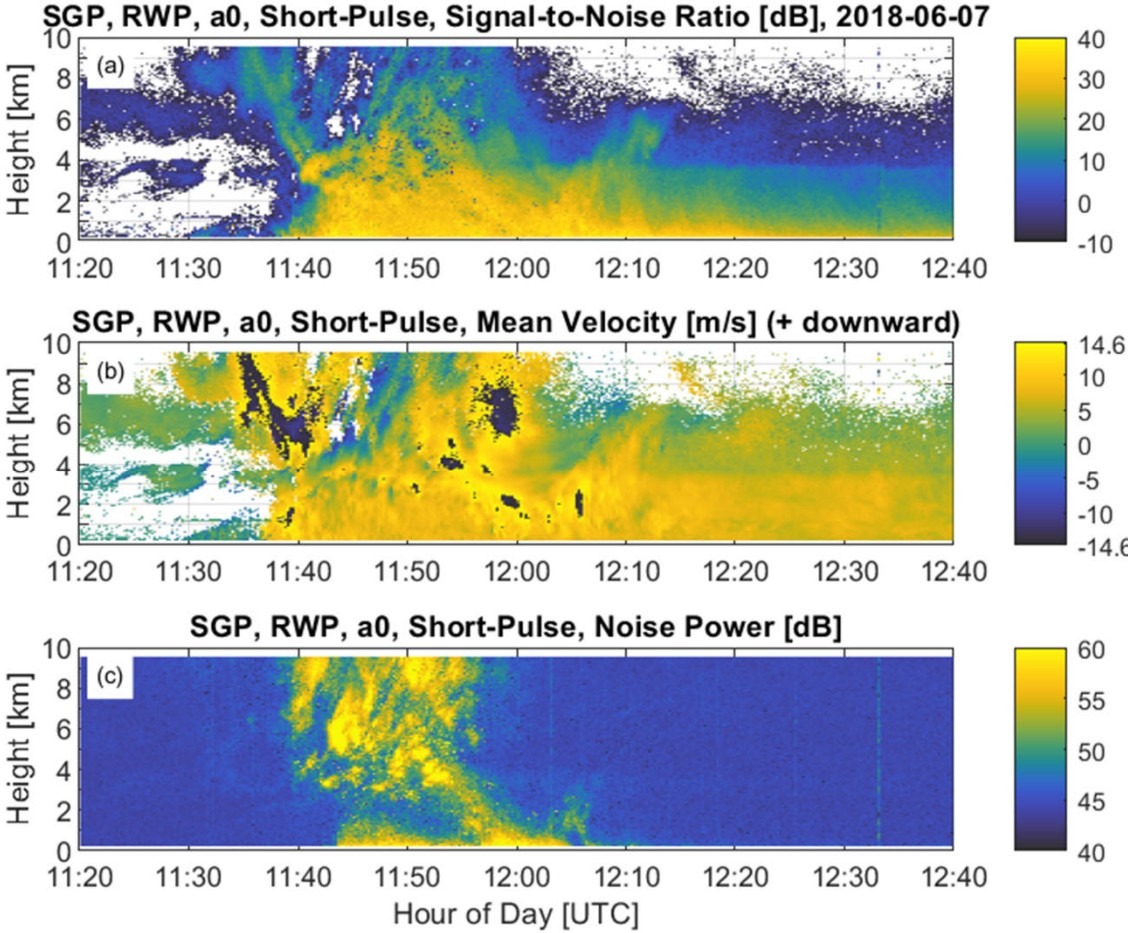

**Figure 1. Radar wind profiler (RWP) spectrum moments calculated with the real-time processing algorithms and downloaded from the DOE ARM archive. RWP is located at the SGP Central Facility. Observations are from the vertically pointing beam using the precipitation short-pulse mode on 07-June-2018 between 11:20 to 12:40 UTC. (a) Signal-to-Noise Ratio (SNR) [dB], (b) mean radial velocity with positive values moving downward toward the radar [m s⁻¹], and (c) spectrum noise power [dB].**

### 3.1 Doppler Velocity Power Spectrum Adjustments and Calculating Spectrum Moments

This subsection describes three processing steps: 1) spectrum adjustments due to Nyquist velocity aliasing, 2) spectrum adjustments due to coherent integration, and 3) recalculating the spectrum moments. Appendix A presents a flow diagram illustrating how these processing steps are applied to a profile of radar observations.

### 3.1.1 Eliminating Nyquist Velocity Aliasing

Nyquist velocity aliasing is when the target radial velocity exceeds the Nyquist velocity and the target appears to be moving in the opposite direction. One velocity aliasing mitigation technique is to concatenate two copies of the same Doppler velocity spectrum to remove the artificial boundary at the Nyquist velocity (Williams et al., 2018). Figure 2 shows an example of

velocity aliasing between 5 and 8 km using precipitation short-pulse mode Doppler velocity power spectra for a single profile collected on 7-June-2018 at 11:58:20 UTC. The original power spectra are plotted within the Nyquist velocity ($V_{Nyquist}$) range of $\pm14.6$ m s$^{-1}$, with downward motions having positive values consistent with raindrop gravitational fall speeds. The original spectra are copied in Fig. 2 to visualize and to mitigate Nyquist velocity aliasing. Specifically, the original downward motions between 0 and 14.6 m s$^{-1}$ are copied to upward motions between -29.2 to -14.6 m s$^{-1}$. The original upward motions between -14.6 and 0 m s$^{-1}$ are copied to downward motions between 14.6 to 29.2 m s$^{-1}$. The red circles in Fig. 2 designate real-time mean radial velocity moments $V_{mean}^{a0}$. Note the jump in $V_{mean}^{a0}$ near 5.5 km from downward to upward motion, which is due to the assumption in the real-time signal processing routines that all signal power is within the Nyquist interval of $\pm V_{Nyquist}$.

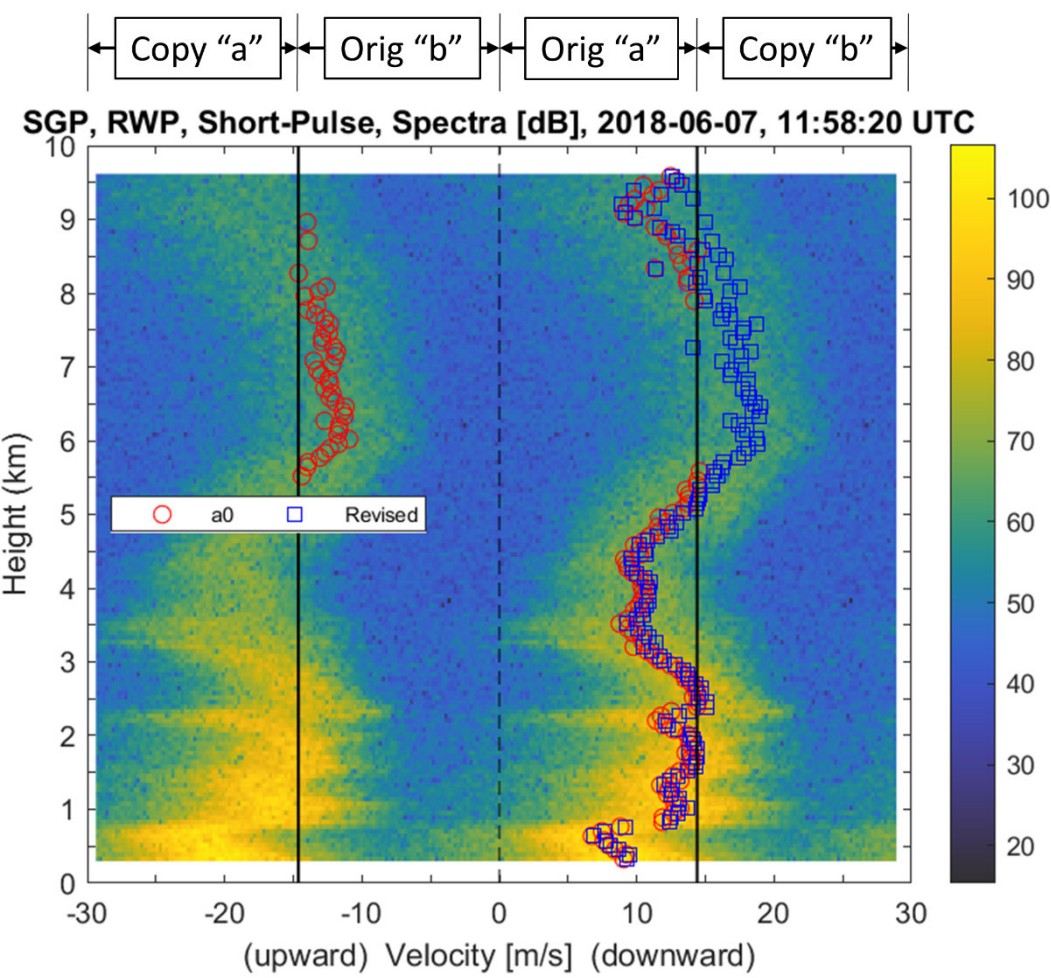

**Figure 2. Spectra profile at time 11:58:20 [UTC] on 7-June-2018. Downward velocities have positive values and are approaching the ground-based radar. Original spectra are plotted between Nyquist velocities -14.6 and 14.6 m s$^{-1}$ and are indicated with solid lines. The portion of original spectra with downward motion is copied to be more upward than the Nyquist velocity (i.e., portion labelled "a") and the portion of original spectra with upward motion is copied to be more downward than the Nyquist velocity (i.e., portion labelled "b"). Red circles designate real-time estimated mean radial velocities and blue squares denote revised mean radial velocities.**

**Dashed line indicates 0 m s⁻¹ velocities. Spectra magnitudes are uncalibrated spectral power density units expressed in decibels (i.e., $10log[S(v)]$ with units dB).**

For spectra that have velocity aliasing, the SNR is biased low when using the assumption that all of the signal power is within $\pm V_{Nyquist}$ . This issue can be visualized in Fig. 3 and shows individual spectra at 6 km (Fig. 3a) and 3 km (Fig. 3b). The signal-to-noise ratio can be estimated using (Riddle et al., 2012):

$$SNR = 10log\left[\frac{\sum_{v_{start}}^{v_{end}}[S(v_i)-\bar{n}]\Delta v}{\bar{n}N_{pts}\Delta v}\right] \qquad \text{[dB]} \qquad\qquad\qquad (1)$$

where $v_{start}$ and $v_{end}$ [m s⁻¹] are the integration limits indicating the start and end velocities of the power spectrum $S(v_i)$

containing signal power [uncalibrated power per (m s⁻¹)], $v_i$ is the velocity bin, $\Delta v$ [m s⁻¹] is the velocity bin resolution, $\bar{n}$ is the spectrum mean noise level [uncalibrated power per (m s⁻¹)] (Hildebrand and Sekhon, 1974), and $N_{pts}$ is the number of points in the spectrum. The real-time processing routine uses only the spectrum between $\pm V_{Nyquist}$ to determine the spectrum moments. In Fig. 3b, the maximum magnitude is near 10 m s⁻¹ (downward), the $v_{start}$ integration limit is near 0 m s⁻¹ and the $v_{end}$ limit stops at the Nyquist velocity of 14.6 m s⁻¹. The spectrum between these integration limits is shaded red and labelled

*a0 spectrum* in Fig. 3. The revised processing routine uses the extended spectrum that spans between $\pm 2V_{Nyquist}$. Since the original spectrum is copied into the extended spectrum, the maximum magnitude peak that occurs near 10 m s⁻¹ (downward) also occurs near -19 m s⁻¹ (upward). The revised processing routine uses information from the previous range gate to select which of the two peaks to process. Appendix A describes the processing steps using a prior velocity $V_{prior}$ to select one of the two peaks. After a peak is selected, the revised processing routine uses the same search technique as the real-time processing

routine, except it uses the extended spectrum illustrated in Figs. 2 and 3. For spectrum shown in Fig. 3b , the $v_{start}$ integration limit is the same determined from the real-time processing routine, but the $v_{end}$ limit extends past the Nyquist velocity and ends where the spectrum crosses the mean noise level near 20 m s⁻¹ (downward). The different integration limits cause the real-time processing method to underestimate both the SNR and mean radial velocity relative to the dealiased method by 0.2 dB and 0.2 m s⁻¹, respectively. As will be seen in the next section, including the incoherent averaging filtering effects will

increase these differences.

In Fig. 2, between 5.5 and 9 km, $V_{mean}^{a0}$ appears to have upward motion. This is because the true maximum spectrum magnitude has a downward velocity occurring outside the $\pm V_{Nyquist}$ boundaries and the aliased peak has an upward velocity. Figure 3a shows the velocity power spectrum at 6 km and the real-time processing routine found integration limits that bound the upward spectrum maximum magnitude peak near -12 m s⁻¹. The integration limits are -14.6 (at the Nyquist boundary) and

approximately -5 m s⁻¹. This *a0 spectrum* region is shaded red in Fig. 3a. In contrast, the revised processing routine selected the downward moving peak in the dealiased spectrum and found integration limits of approximately 11 and 24 m s⁻¹ downward (spectrum region with blue strips). The different integration limits produce significantly different mean radial velocities of $V_{mean}^{a0}$ equal to -10.5 m s⁻¹ and $V_{mean}^{revised}$ equal to 17.9 m s⁻¹.

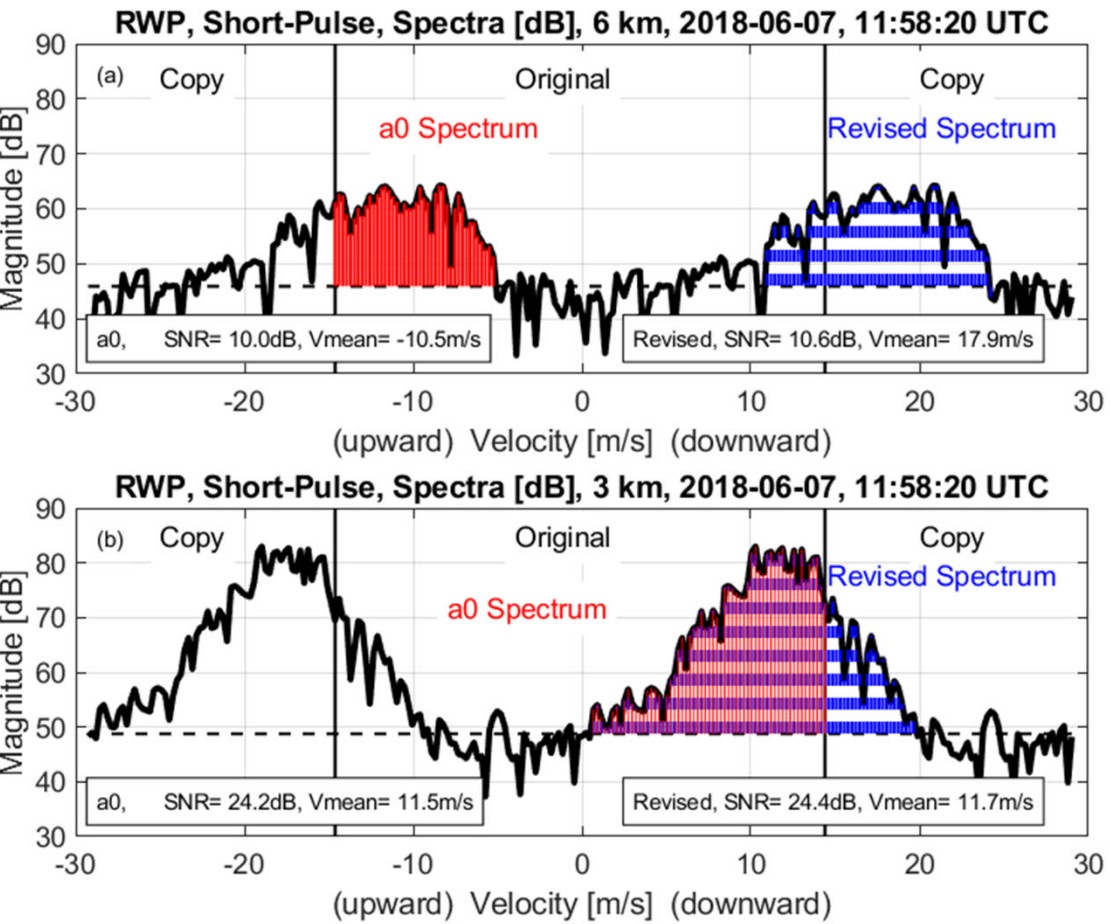

Figure 3. Example of integration limits used in the real-time and revised spectrum moment estimation algorithms. Uncalibrated spectral power density expressed in decibels [dB] for profile at 11:58:20 [UTC] on 7-June-2018 at (a) 6 km and (b) 3 km. Red shading and blue horizontal bars indicate spectral power density used to estimate *a0* and revised moments, respectively.

### 3.1.2 Coherent Integration Adjustment

Coherent integration is a signal processing technique that accumulates the radar measured in-phase and quadrature voltages (aka, *I* and *Q* voltages) over consecutive transmitted pulses. Sinusoidal oscillations with slowly varying phase over the accumulation interval are said to be coherent and their accumulated *I* and *Q* voltages cause an increase in signal power. Conversely, accumulating *I* and *Q* voltages over high frequency oscillations, including noise fluctuations, will produce smaller magnitude accumulated *I* and *Q* voltages resulting in smaller signal power. Thus, coherent integration increases radar detection by acting as a low-pass filter that increases low-frequency signal powers and decreases high-frequency noise power (Farley, 1985).

Coherent integration is also known as time-domain averaging (TDA) and is implemented by changing the number of coherent integration samples $N_{coh}$, which changes the effective time between transmitted samples and decreases the Nyquist velocity using:

$$V_{Nyquist} = \left(\frac{\lambda}{4}\right)\left(\frac{1}{N_{coh}T_{IPP}}\right) \tag{2}$$

where $\lambda$ is the radar operating wavelength and $T_{IPP}$ is the inter-pulse period (aka, time between transmitted pulses). Coherent integration also applies a boxcar filter to the $I$ and $Q$ voltage time-series samples before integrating, which is equivalent to applying a low-pass filter to the integrated time-series (Wilfong et al., 1999). Since coherent integration is performed before computing the FFT on the complex $I$ and $Q$ voltage samples, the low-pass filter manifests as a reduction in FFT signal power magnitude as a function of velocity $v_i$ and has the form (Schmidt et al., 1979):

$$S_{recorded}^{signal}(v_i) = S_{expected}^{signal}(v_i)\left[\frac{sin^2\left[\frac{\pi\left(\frac{v_i}{\Delta v}\right)}{N_{pts}}\right]}{(N_{coh}^2)\left(sin^2\left[\frac{\pi\left(\frac{v_i}{\Delta v}\right)}{N_{coh}N_{pts}}\right]\right)}\right] \tag{3}$$

where $S_{recorded}^{signal}(v_i)$ is the recorded signal power spectrum at velocity bin $v_i$, $S_{expected}^{signal}(v_i)$ is the expected signal power spectrum without any time-domain low-pass filtering effects, and $N_{pts}$ is the number of complex $I$ and $Q$ samples after coherent integration, which is also the number of velocity bins in the power spectrum after performing the FFT calculation. The ratio $\left(\frac{v_i}{\Delta v}\right)$ yields integers from $\frac{-N_{pts}}{2}$ to $\frac{N_{pts}}{2}$. Note that the low-pass filter response function (the expression within the

280 square brackets in (3)) has a magnitude of one when $v_i = 0$ and decreases with increasing $v_i$.

The impact of the TDA low-pass filter can be mitigated by applying a correction factor to the recorded Doppler velocity power spectra as discussed in Wilfong et al. (1999). Since the low-pass filter only affects coherent signals, the correction factor should only be applied to the signal portion of the power spectrum and not to the random noise power. Thus, the TDA corrected power spectrum $S_{TDA}(v_i)$ is estimated using:

$$S_{TDA}(v_i) = [S(v_i) - \bar{n}]\left[\frac{(N_{coh}^2)\left(sin^2\left[\frac{\pi\left(\frac{v_i}{\Delta v}\right)}{N_{coh}N_{pts}}\right]\right)}{sin^2\left[\frac{\pi\left(\frac{v_i}{\Delta v}\right)}{N_{pts}}\right]}\right] + \bar{n} \tag{4}$$

where $S(v_i)$ is the recorded Doppler velocity power spectrum. For the precipitation short-pulse mode, the correction factor magnitude (the expression in the square brackets in (4)) at $\pm V_{Nyquist}$ is 2.47 in natural units as in equation (4) or 3.9 dB in decibels. Figure 4 shows the recorded power spectra shown in Fig. 3 with the revised spectrum corrected for the TDA filtering expressed in (4). The $SNR$ and mean radial velocity moments for real-time moments and the revised spectrum are listed in Fig.

4. Comparing the non-TDA and TDA corrected moments for the dealiased spectra at 6 km (see Figs. 3a and 4a, respectively), indicates the SNR increased 7.4 dB and the mean radial velocity became more downward by 1.6 m s⁻¹ when including the TDA filter correction. Note that the difference in $a0$ and TDA corrected mean radial velocities at 6 km is 30 m s⁻¹ (see Fig. 4a)

and is not a multiple of $\pm 2V_{Nyquist}$ ($\pm 29.2$ m s-1). This indicates that simple integer $\pm 2V_{Nyquist}$ adjustments, as proposed by Tridon et al. (2013), will not account for improper integration limits used in the real-time processing routines.

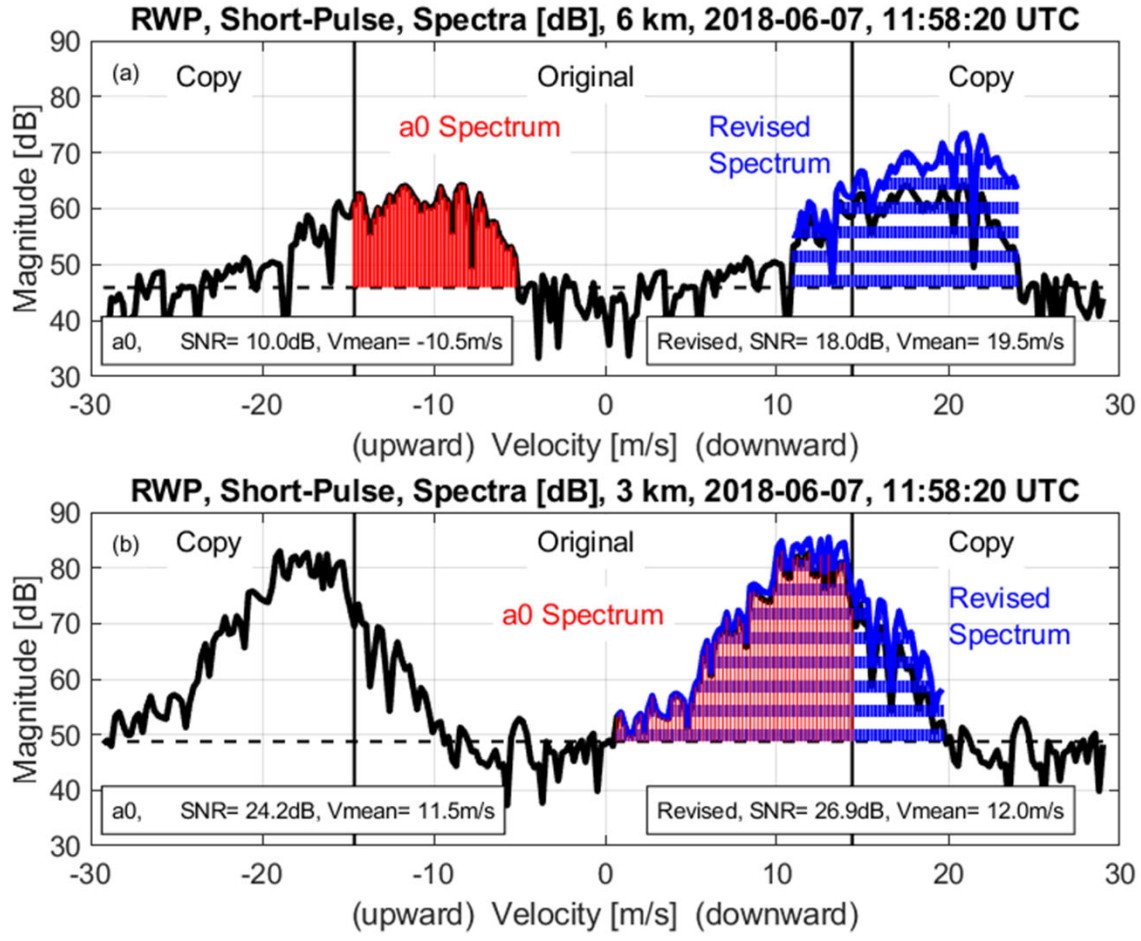

Figure 4. Similar to Fig. 3, except the revised spectrum (blue line and blue horizontal bars) have been TDA corrected using (4).

### 3.1.3 Calculating Spectrum Moments

After adjusting the recorded spectrum due to Nyquist velocity aliasing and coherent integration effects, the spectrum moments are calculated following the method and equations presented in Williams et al. (2018) Appendix A. The calculated revised spectrum moments include spectrum signal power ($P_{signal}^{revised}$) [dB], spectrum noise power ($P_{noise}^{revised}$) [dB], signal-to-noise ratio ($SNR^{revised}$) [dB], spectrum mean radial velocity ($V_{mean}^{revised}$) [m s⁻¹], spectrum standard deviation ($\sigma^{revised}$) [m s⁻¹], spectrum width ($W^{revised} = 2\sigma^{revised}$) [m s⁻¹], spectrum skewness, and spectrum kurtosis. The dealiasing procedure described in Section 3.1.1 produces a spectrum with two peaks (e.g., see Figs. 2 and 3). To determine which peak to analyse, the processing

routine starts at the lowest range gate and calculates a prior velocity $V_{prior}$ that is used to select a peak in the next range gate.

More details of the processing steps are provided in Appendix A.

### 3.2 Signal-to-Noise Ratio (SNR) Adjustment

Signal power is estimated relative to the estimated mean noise power and is quantified with the signal-to-noise ratio $SNR$. If the noise power estimate is too large, then the signal-to-noise ratio and the inferred signal power are underestimated. The *a0* processed noise power $P_{noise}^{a0}$ shown in Fig. 1c had increased magnitudes at nearly all range gates during the convective rain

event between approximately 11:35 and 12:10 UTC. This increased noise power is not expected for RWPs because the gain is constant with range so that noise power should be independent of range. Also, Fig. 2 shows signal power spread over a large fraction of the velocity spectrum. These two features are linked: the broad signal spectra are causing increased noise power estimates. Specifically, as the signal velocity power spectrum broadens and occupies more of the velocity spectrum, the noise estimator is biased by the inclusion of signal power. The RWP online signal processing uses the Hildebrand and Sekhon (1974)

noise level estimator to separate noise-only spectral bins from signal-plus-noise spectral bins based on the statistical properties of both populations (for more details see Merritt, 1995 and Wilfong et al., 1999). If the signal-plus-noise spectral bins are included in the noise-only population, then the noise level estimate will be biased high leading to an underestimated $SNR$. To correct for this low $SNR$ bias, a reference noise power $P_{noise}^{reference}$ [dB] is determined and an adjusted SNR is estimated using:

$$SNR_{adjusted}^{revised} = SNR^{revised} + P_{noise}^{revised} - P_{noise}^{reference} \qquad [dB] \qquad (5)$$

where $SNR^{revised}$ and $P_{noise}^{revised}$ are moments calculated in Section 3.1.3.

    The noise power for every spectrum is estimated using the method outlined in Hildebrand and Sekhon (1974). The reference noise power $P_{noise}^{reference}$ is the median noise power derived from all spectra collected on a given day. Figure 5 shows the daily median noise power for the precipitation short-pulse (black plusses) and long-pulse (red crosses) for the 8-year dataset. The jump in daily median noise power in mid-2017 corresponds to replacing the transmitter with a used, yet updated

version, from the same RWP vendor. It is interesting to note that the seasonal noise variation decreased with the updated transmitter and not when the equipment shelter air conditioning system was updated in mid-2016.

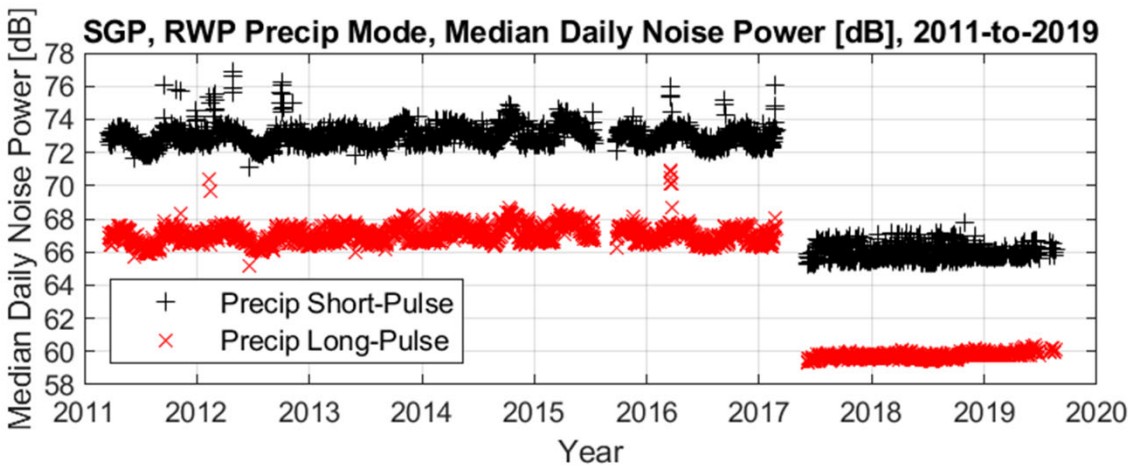

**Figure 5. Daily median noise level for the precipitation short-pulse (black plusses) and long-pulse (red crosses) mode for observations between 2011 and 2019.**

Figure 6 illustrates the impact of adjusting the signal-to-noise ratio with the reference noise power. Fig. 6a shows the real-time estimated $SNR^{a0}$ (thick line) and $P_{noise}^{a0}$ (thin line) profiles. The large variations in $P_{noise}^{a0}$ between 4 and 5 km appear as large and inverse variations in $SNR^{a0}$. Figure 6b shows the adjusted signal-to-noise ratio using two methods. The method described in Tridon et al. (2013) uses the real-time moments ($SNR^{a0}$ and $P_{noise}^{a0}$) to estimate the adjusted signal-to-noise ratio $SNR_{adjusted}^{a0}$ (thick blue line in Fig. 6b). The method described herein recalculates the moments and then estimates the adjusted

signal-to-noise ratio $SNR_{adjusted}^{revised}$ using equation (5) (thin red line). The profile offset in Fig. 6b is due to different reference noise powers used in the two methods. The $SNR_{adjusted}^{a0}$ has more variability than $SNR_{adjusted}^{revised}$, indicating that the revised spectra reprocessing method produces smoother, more vertically consistent, SNR vertical profiles than the Tridon et al. (2013) method.

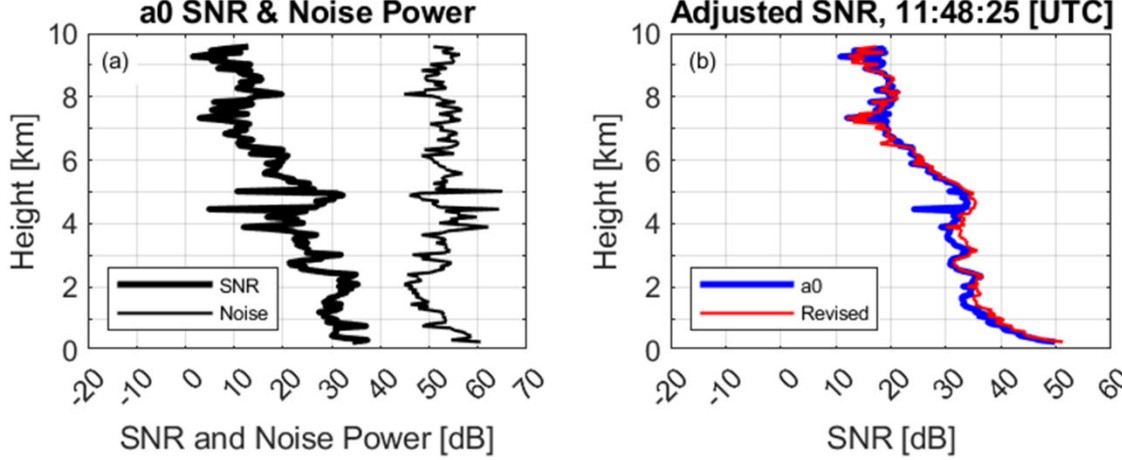

**Figure 6. Moment profiles at time 11:48:25 [UTC] on 7-June-2018. (a) SNR and spectrum noise power from real-time spectrum processing routines. (b) Adjusted SNR using the *a0* moments shown in panel (a) (thick blue line) and adjusted SNR using the revised spectral method (thin red line). The adjusted SNR profiles are offset because of different reference noise values.**

### 3.3 Calibrating Reference Beam to Surface Disdrometer

The precipitation short-pulse beam is defined as the RWP reference beam and the radar reflectivity factor $Z^{PrecipShort}$ [dBZ] for this beam is estimated from the adjusted signal-to-noise ratio $SNR^{PrecipShort}_{adjusted}$ using

$$Z^{PrecipShort}(r) = SNR^{PrecipShort}_{adjusted} + 20\log(r) + C^{PrecipShort} \qquad \text{[dBZ]} \qquad (6)$$

where $r$ [m] is range from the radar and $C^{PrecipShort}$ [dB] is the calibration constant. To estimate the calibration constant $C^{PrecipShort}$, an initial value of $C^{PrecipShort}$ equal to 0 dB is selected and equation (6) is used to estimate the RWP reflectivity factor at all range gates. These initial RWP reflectivity factors at 500 m above ground level are averaged into 1-minute quantities and then compared with the 1-minute surface disdrometer radar reflectivity factors. Using only disdrometer reflectivity factors between 20-to-40 dBZ, the reflectivity factor differences are calculated for RWP lags between $\pm4$ minutes. Both positive and negative lags are needed because the two instruments are separated by approximately 100 m and the horizontal wind speed and direction can cause the surface rain observations to occur before the radar observations at 500 m altitude. Figure 7 shows scatter plots and statistics of mean, standard deviation, and Pearson's correlation coefficient for the 7-June-2018 rain event at nine different lags. For this rain event, the distribution in Fig. 7d is selected for calibration because it has the highest Pearson's correlation coefficient of 0.95.

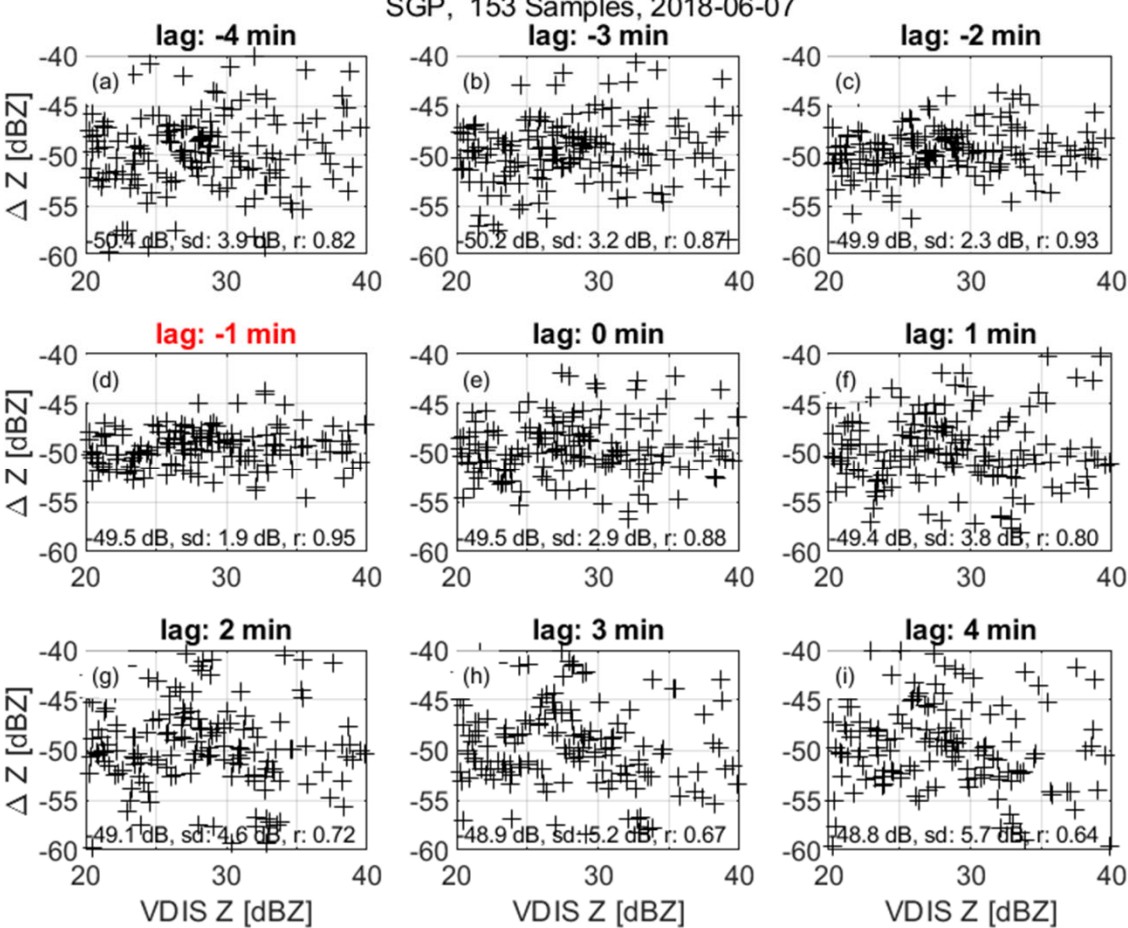

Figure 7. Scatter plots of reflectivity factor differences between RWP precipitation short-pulse mode (with $C^{PrecipShort} = 0$ dB) and surface disdrometer for different minute lags for the rain event on 7-June-2018. Positive lags indicate RWP shifted to later time. Lag times for each panel are (a) -4 min (b) -3 min, (c) -2 min, (d) -1 min (e) 0 min, (f) 1 min, (g) 2 min (h) 3 min, and (i) 4 min. This rain event had 153 minute samples with surface disdrometer reflectivity factor between 20 and 40 dBZ. Each panel indicates rain event mean difference, standard deviation (sd), and Pearson's correlation coefficient (r). Panel (d) has the largest Pearson's correlation coefficient and is used for calibrating this event. The calibration constant for this event is $C^{PrecipShort} = -49.5$ dB.

Using the calibration constant and lag determined from Fig. 7d (i.e., $C^{PrecipShort} = $-49.5 dB and -1-minute lag), Fig. 8a shows the time-height cross-section of calibrated RWP precipitation short-pulse mode radar reflectivity factor. Figure 8b shows a time-series of RWP reflectivity factor at 500 m (red crosses) and the surface disdrometer reflectivity factor (black plusses). The blue thin lines in Fig. 8b at 20 and 40 dBZ indicate the reflectivity factor range used for calculating the RWP and disdrometer differences, which are shown in Fig. 8c. Also shown in Fig. 8c are the statistics for this lag, including lag, number of samples, calibration constant, standard deviation, and Pearson's correlation coefficient. Figure 8d shows surface disdrometer rain rate $RR$ and mass-weighted mean diameter $D_m$. The standard deviation of 1.9 dB for this event is due to spatiotemporal mismatch between the surface disdrometer and radar sample volume as well as measurement uncertainties of

both instruments, and is comparable to 1-to-2 dB measurement uncertainties of side-by-side surface disdrometers (Tapiador et

al., 2017; Wang et al., 2021). Note that the lag is only used in the calibration procedure and not used as a time offset for any

other purpose.

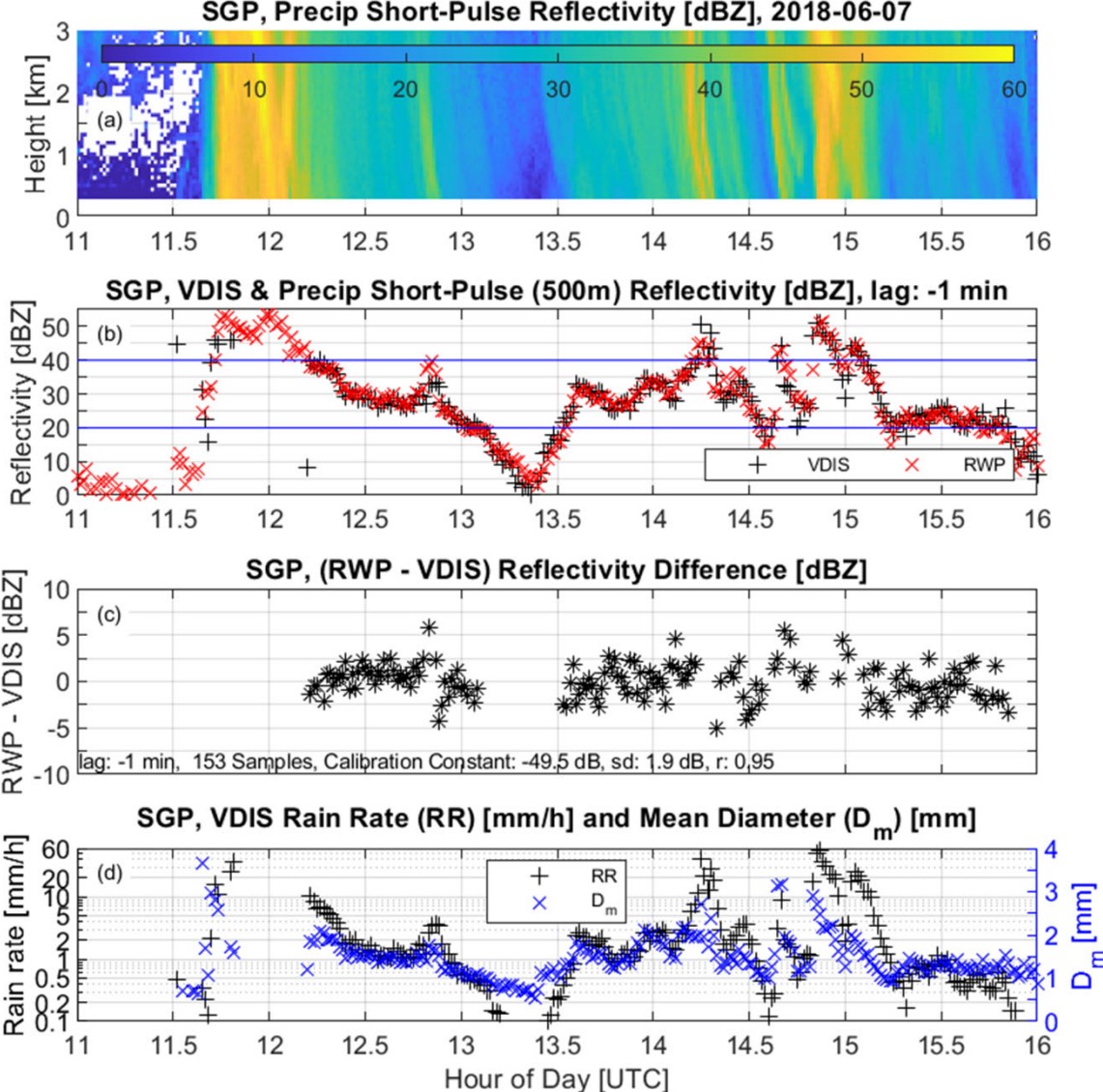

**Figure 8. RWP precipitation short-pulse mode and surface disdrometer observations from 7-June-2018 between 11 and 16 UTC. (a) RWP radar reflectivity factor with calibration constant of -49.5 dB, (b) RWP radar reflectivity factor (red crosses) at 500 m range**

**with -1-minute lag and surface 2DVD radar reflectivity factor (black plusses), (c) reflectivity factor difference (RWP – VDIS) for samples with VDIS reflectivity factor within 20 to 40 dBZ as indicated with blue thin lines in panel (b), and (d) disdrometer rain rate $RR$ and mean diameter $D_m$. Statistics of lag, number of samples, calibration constant, standard deviation (sd), and Pearson's correlation coefficient (r) are shown in panel (c).**

Figure 9 shows improved moments and calibrated reflectivity factors for the same rain event shown in Fig. 1. The top
panel (Fig. 9a) shows the revised adjusted signal-to-noise ratio ($SNR_{adjusted}^{PrecipShort}$) and the middle panel (Fig. 9b) shows the
revised mean radial velocity ($V_{mean}^{PrecipShort}$). Compared to the *a0* real-time processed moments, the reprocessed moments shown
in Figs. 9a and 9b show improved data quality and uniformity. The calibrated radar reflectivity is shown in Fig. 9c.

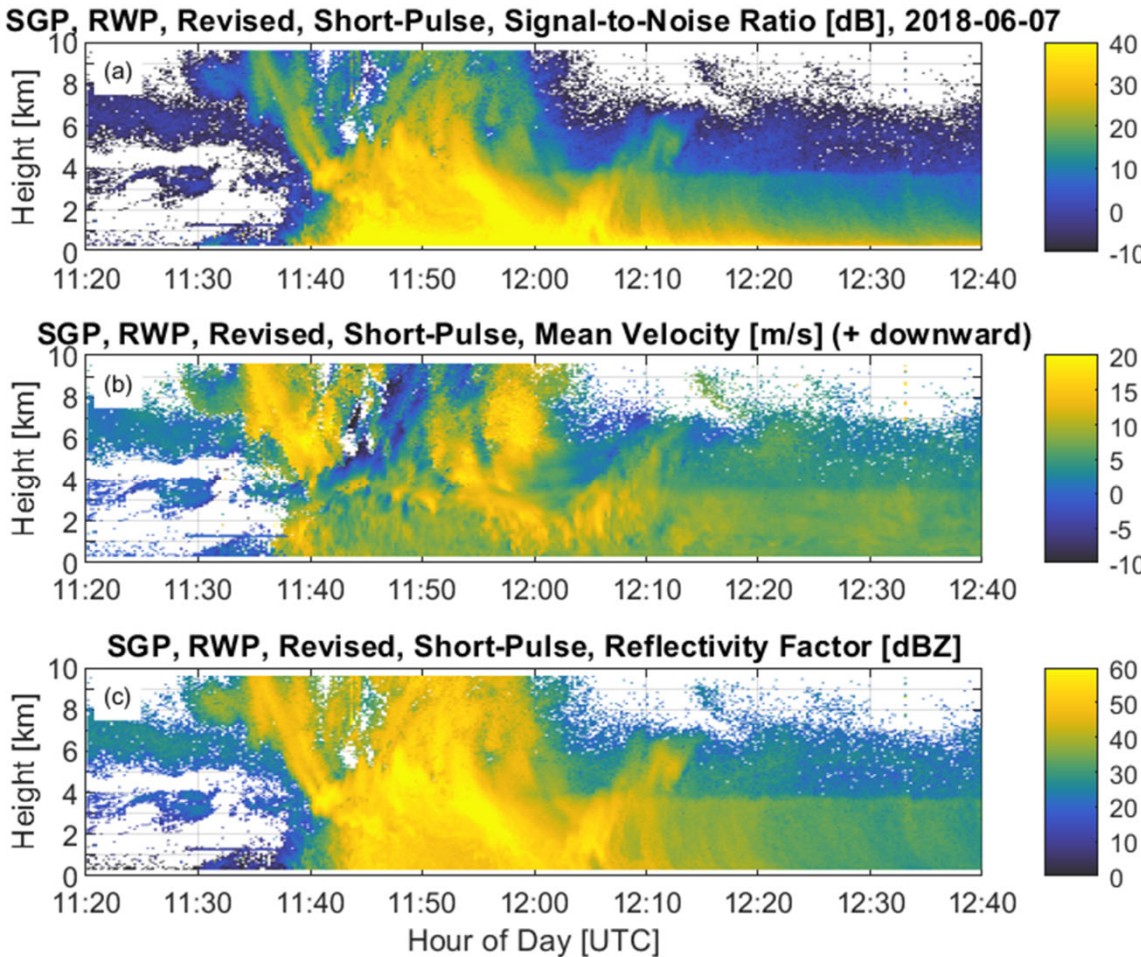

**Figure 9. Similar to Fig. 1 except RWP spectrum moments for the precipitation short-pulse mode calculated with the revised**
**processing algorithms. (a) Signal-to-noise ratio $SNR_{adjusted}^{PrecipShort}$ [dB], (b) mean radial velocity $V_{mean}^{PrecipShort}$ [m s⁻¹] with positive values**
**moving downward consistent with raindrop gravitation fall speeds, and (c) surface disdrometer calibrated radar reflectivity factor**
$Z^{PrecipShort}$ **[dBZ].**

### 3.4 Relative Calibration Constants for Other Radar Beams

The radar sensitivity can be adjusted by changing the transmitted pulse length, the number of coherent integrations, and the
number of averaged Doppler velocity spectra. Using the precipitation short-pulse mode as the reference beam, the expected
relative change in sensitivity for the other four radar beams can be estimated using:

$$C_{relative}^{OtherMode} = 20log\left[\frac{\Delta R^{MUT}}{\Delta R^{PrecipShort}}\right] + 10log\left[\frac{N_{coh}^{MUT}}{N_{coh}^{PrecipShort}}\right] + 5log\left[\frac{N_{spc}^{MUT}}{N_{spc}^{PrecipShort}}\right] + 20log[sin(\theta_{el})] \qquad (7)$$

where $\Delta R$ is the range resolution, $N_{coh}$ is the number of coherent samples, $N_{spc}$ is the number of averaged power spectra, $\theta_{el}$ is the elevation angle from the horizon, and the superscripts *PrecipShort* and *MUT* represent the precipitation short-pulse mode and the mode under test (MUT), respectively. Using the values from Table 1 and equation (7), Table 3 lists the expected relative sensitivities for the precipitation long-pulse mode and the wind mode. The last term in (7) represents the decrease in gain associated with beam pointing direction in phased array antennas (Balanis, 1997). As the beam pointing direction deviates from broadside (aka, vertical direction in the RWP), the projected antenna area decreases causing the gain to decrease and beam width to increase (Balanis, 1997; Palmer et al., 2022). System losses and variations in antenna gain cause the measured relative sensitivities to deviate from the expected values listed in Table 3.

The reflectivity factor for the other four radar beams follows equation (6) with the addition of the relative calibration constant $C_{relative}^{OtherMode}$ [dB] and is estimated using:

$$Z^{OtherMode}(r) = SNR_{adjusted}^{OtherMode} + 20\log(r) + \left(C^{PrecipShort} - C_{relative}^{OtherMode}\right) \qquad [dBZ]. \qquad (8)$$

The negative sign in the bracketed term is because a positive $C_{relative}^{OtherMode}$ indicates this mode is more sensitive than the precipitation short-pulse mode and will produce a larger $SNR_{adjusted}^{OtherMode}$ for the same radar reflectivity factor. Note that weaker radar reflectivity factors will be detected at further ranges at the expense of possible receiver saturation from large reflectivity factor targets at close range.

**Table 3. Expected relative sensitivity of other radar beams compared with the reference precipitation short-pulse beam. Relative sensitivity has three terms in equation (7) and is dependent on range resolution $\Delta R$, coherent integration $N_{coh}$, and number of averaged Doppler velocity power spectra $N_{spc}$.**

| Relative Sensitivity | | Precipitation Mode | Wind Mode | | |
| --- | --- | --- | --- | --- | --- |
| | | Long-Pulse | BeamV | BeamA | BeamB |
| Elevation angle [degree] | | 90° | 90° | 77° | 77° |
| Azimuth angle [degree] | | 22° | 22° | 22° | 292° |
| $20log\left[\frac{\Delta R^{MUT}}{\Delta R^{precipShort}}\right]$ | [dB] | 16.5 | 4.6 | 4.6 | 4.6 |
| $10log\left[\frac{N_{coh}^{MUT}}{N_{coh}^{PrecipShort}}\right]$ | [dB] | -2.2 | 5.5 | 5.5 | 5.5 |
| $5log\left[\frac{N_{spc}^{MUT}}{N_{spc}^{PrecipShort}}\right]$ | [dB] | 0.6 | 3.0 | 3.0 | 3.0 |
| $20log[sin(\theta_{el})]$ | [dB] | 0.0 | 0.0 | -0.2 | -0.2 |
| $C_{relative}^{OtherMode}$ | [dB] | 14.9 | 13.1 | 12.9 | 12.9 |

To estimate relative sensitivities between the other beams and the reference beam, reflectivity factors are estimated at all profiles and range gates using equation (8) with $C_{relative}^{OtherMode}$ set to zero, and then estimating the differences from nearby precipitation short-pulse mode observations. Figure 10 shows scatter plots and histograms of reflectivity factor differences for the precipitation long-pulse beam during the 7-June-2018 rain event. Valid observations are constrained to be within the height interval of 800 and 2100 m and precipitation short-pulse reflectivity factors greater than 30 dBZ. Over 13,000 valid samples are used from this event to calibrate the precipitation long-pulse beam. The mean relative offset is 15.5 dB for this event, with a standard deviation of 1.3 dB. The relative calibration constant $C_{relative}^{PrecipLong}$ is set to 15.5 dB and implies that the long-pulse mode is more sensitive and produces a larger signal-to-noise ratio for the same radar reflectivity factor as expressed in equation (8).

Figure 11a and 11b show the time-height cross-sections of cross-calibrated precipitation short- and long-pulse reflectivity factors at their native resolution for the 7-June-2018 rain event. Figure 11c shows the precipitation long-pulse relative calibration offset for each matched short- and long-pulse observation. The relative calibration offsets shown in Fig. 11c are the same samples used to produce Fig. 10 and indicate the limited height interval used in the comparison to avoid large reflectivity gradients near the radar bright band caused by melting particles.

Estimating the relative calibration offsets for the three wind beams follows the same procedure used for estimating the precipitation long-pulse beam relative calibration offset. As expected, the calibration offsets for the oblique beams have more event-to-event variability than the vertically pointing wind mode beam and will be discussed further in the next section and shown in Fig. 14.

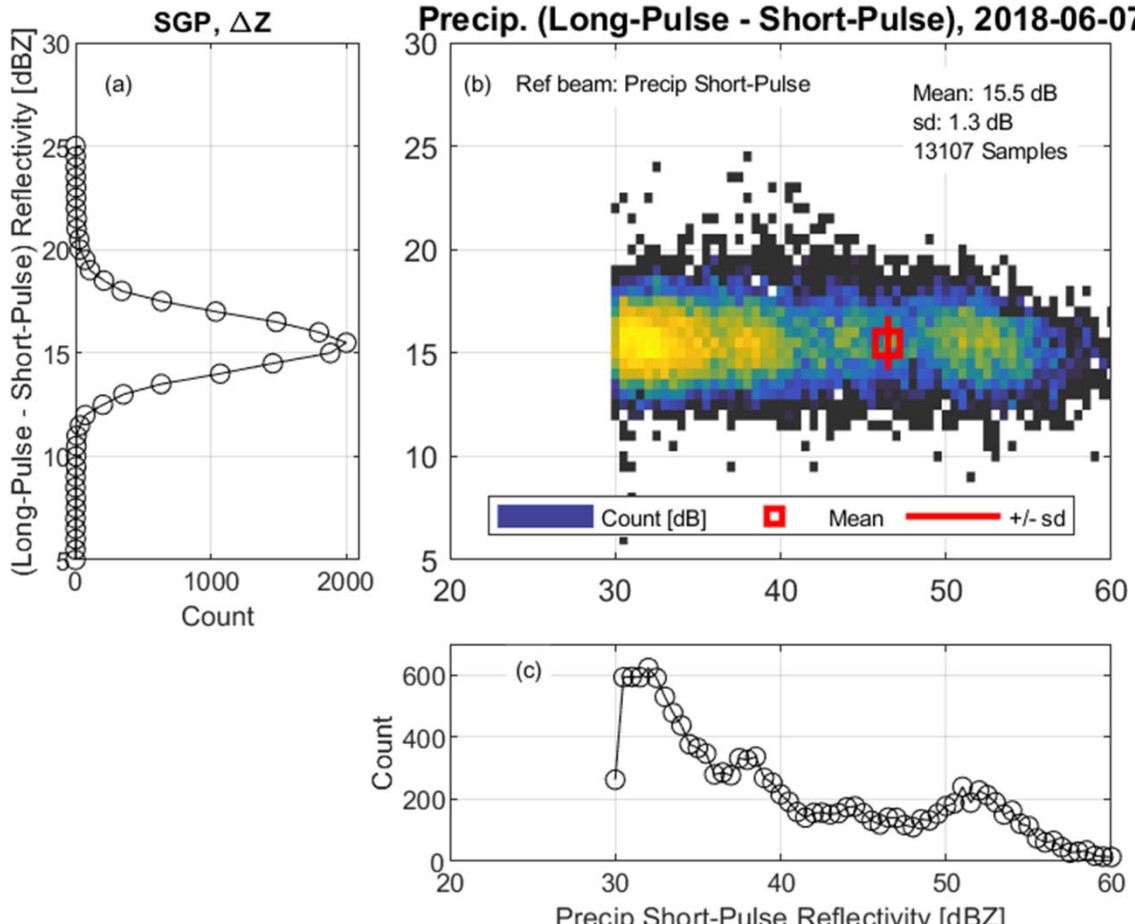

**Figure 10. Reflectivity factor differences between precipitation long-pulse beam with $C_{relative}^{PrecipLong} = 0$ [dB] and disdrometer calibrated precipitation short-pulse beam observations for the rain event on 7-June-2018. Observations are limited to heights between 800 and 2100 m and precipitation short-pulse beam reflectivity greater than 30 dBZ. (a) Histogram of reflectivity difference (long-pulse − short-pulse) and indicates relative calibration offset, (b) relative 2-dimensional count of reflectivity difference, (c)**
**histogram of disdrometer calibrated precipitation short-pulse reflectivity.**

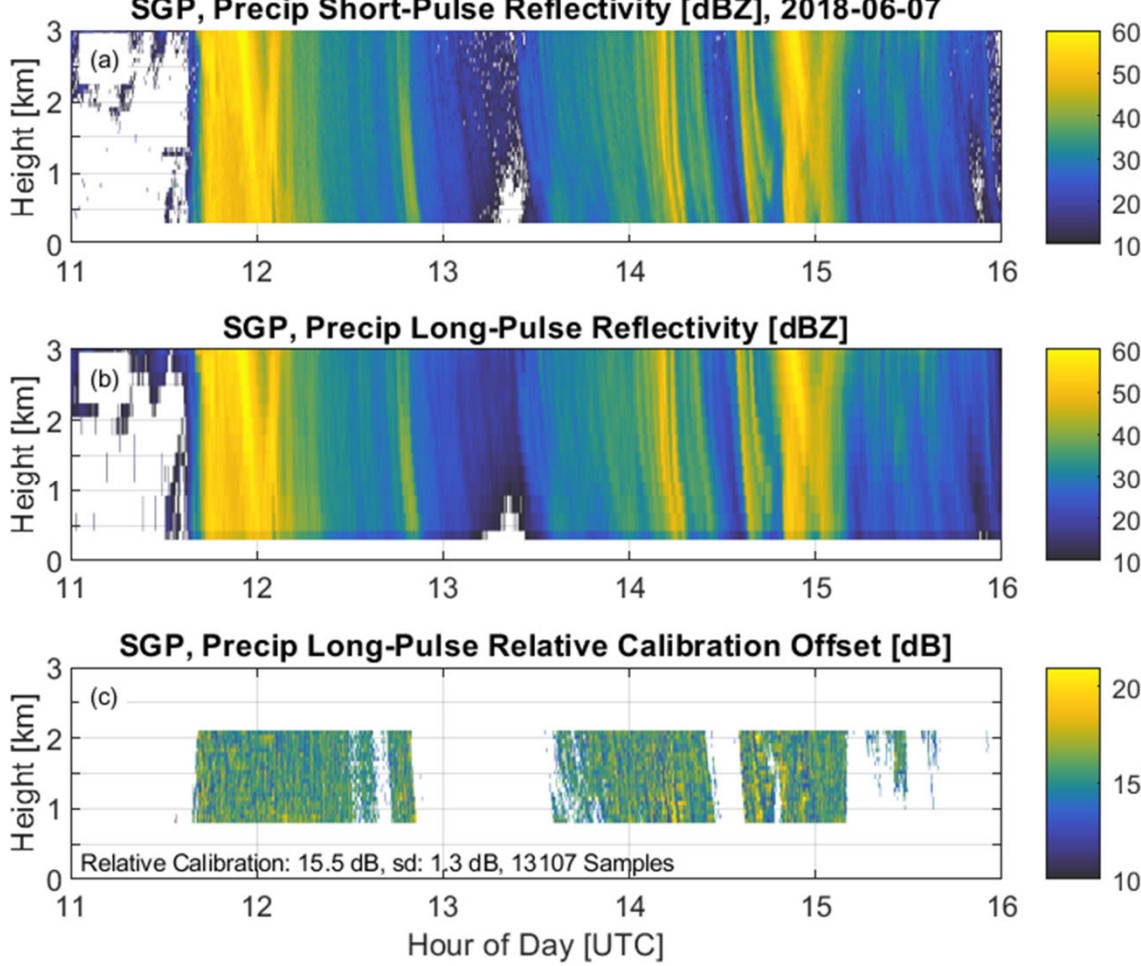

**Figure 11. Time-height cross-sections for the 7-June-2018 rain event. (a) Surface disdrometer calibrated precipitation short-pulse reflectivity factor $Z^{PrecipShort}$, (b) cross-calibrated precipitation long-pulse reflectivity factor $Z^{PrecipLong}$ with $C_{relative}^{PrecipLong} = 15.5$ dB, and (c) the precipitation long-pulse relative calibration offset for each matched short- and long-pulse observation. Relative calibration offset is only calculated for $Z^{PrecipShort} > 30$ dBZ and for heights between 800 and 2100 m.**

## 4 Results

This section explores how the individual rain event precipitation short-pulse beam calibration constants varied over the 8-year record from 22-March-2011 to 18-August-2019. The variation of the relative calibration constants is examined as a function of ageing hardware and a function of changing radar hardware after equipment failures.

### 4.1 Reference Beam Calibration: Event, Monthly, and 3-Month Intervals

From 22-March-2011 to 18-August-2019, the precipitation short-pulse beam calibration constant $C^{PrecipShort}$ was estimated on 340 days, each having at least 120 minutes of surface disdrometer reflectivity factor greater than 20 dBZ. Figure 12 shows

$C^{PrecipShort}$ for every valid precipitation event using black plus symbols. The calibration constant is approximately -50 dB at the beginning of this record in 2011 and then increases to about -35 dB near the beginning of 2015. There is an abrupt drop in calibration constant near the end of 2015, and then the calibration constant steadily increases until the end of this dataset in 2019. Snow events were not included in the calibration procedure.

An increase in calibration constant, without changing operating parameters, indicates the radar sensitivity is degrading. Referring to equation (6), if the reflectivity factor is constant and the measured SNR decreases because of ageing radar hardware, then the calibration constant must increase. Thus, from early 2011 to mid-2015, the calibration was stable until early 2013, then increased approximate 15 dB over the next 2 years indicating a rapid change in calibration. There was a hardware failure in mid-2015.

The gaps in measurements in mid-2015 and early 2017 are when the radar was not operating. A new antenna phase shifter module was installed in September 2015, and the calibration constant dropped by about 10 dB relative to the old hardware. In mid-2017, a new radar transmitter and receiver module was installed and the mean noise level dropped by about 7 dB (see Fig. 5), but the short-pulse beam calibration constant did not change significantly. The steady increase in calibration constant from 2016 through 2019 suggests an approximate 3 dB/year decrease in sensitivity for this modified radar. Though not documented publicly, similar decreasing sensitivity rates have been estimated in other NOAA UHF wind profilers and have been attributed to delamination of the fibreglass patch antenna (Ecklund et al., 1988).

The slow change in calibration constant between precipitation events suggests that the disdrometer-to-RWP calibration procedure could be performed using fixed time intervals instead of individual rain events. To test this hypothesis, calibration constants were determined using all rain events during 1-month and 3-month intervals (i.e., months of JFM, AMJ, JAS, and OND). The 1- and 3-month calibration constants are plotted in Fig. 12 using blue squares and red triangles, respectively. The blue and red vertical lines represent 1- and 3-month calibration constant standard deviations, with mean standard deviations over the 9-year record equal to 3.6 and 2.9 dB, respectively. These standard deviations represent variations due to spatiotemporal mismatch of surface disdrometer and radar measurements, instantaneous measurement uncertainties of both instruments, as well as aging hardware over the sampling interval.

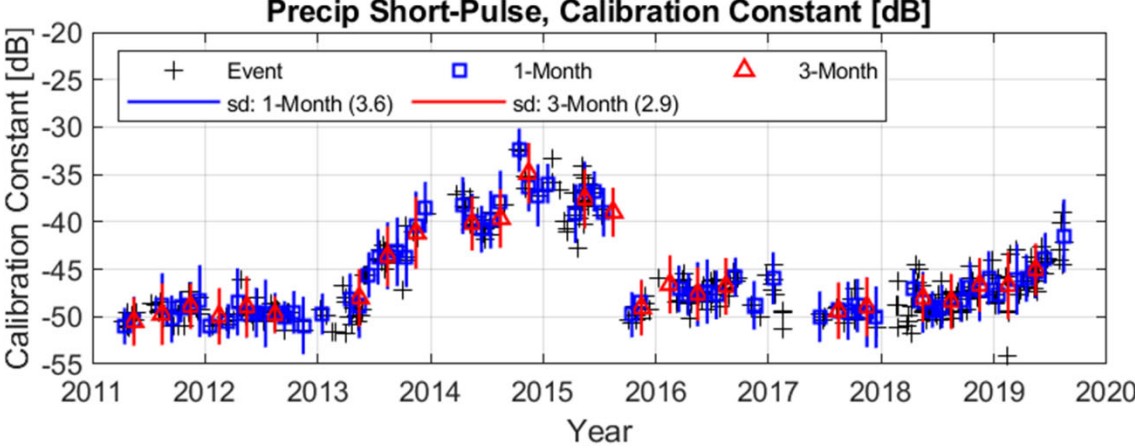

**Figure 12. Precipitation short-pulse beam calibration constant $C^{PrecipShort}$ [dB] from March 2011 through July 2019 estimated using individual rain events (black plusses), 1-month interval (blue squares), and 3-month interval (red triangles). Vertical blue and red lines are +/- standard deviation for 1- and 3-month interval calculations, respectively. Mean standard deviations over the 9-year dataset were 3.6 and 3.0 dB for the 1- and 3-month intervals, respectively.**

## 4.2 Relative Calibration for each Hardware Calibration Period

Since the radar operating parameters did not change during the 2011 to 2019 interval, variations in relative calibration constants will depend on changes to the radar hardware. This section examines how the precipitation long-pulse and wind mode relative calibration constants evolved with hardware changes.

### 4.2.1 Changes in Precipitation Long-Pulse Relative Calibration Constant

The relative calibration constants for the precipitation long-pulse beam were estimated for every day with at least 1000 precipitation short- and long-pulse range gate samples between 800 and 2100 m range and with precipitation short-pulse reflectivity factor greater than 30 dBZ. The lower height limit of 800 m is to ensure the long-pulse beam observations are beyond the radar blind zone, and the 2100 m limit is to avoid reflectivity factor gradients near the melting layer. The precipitation long-pulse relative calibration constant was estimated for the 690 days meeting these criteria and are shown in Fig. 13 using black crosses. The dashed lines are the mean relative calibration values for each stable hardware interval labelled A through E (see Table 2). The relative calibration constant mean and standard deviation for each interval are listed in Table 4.

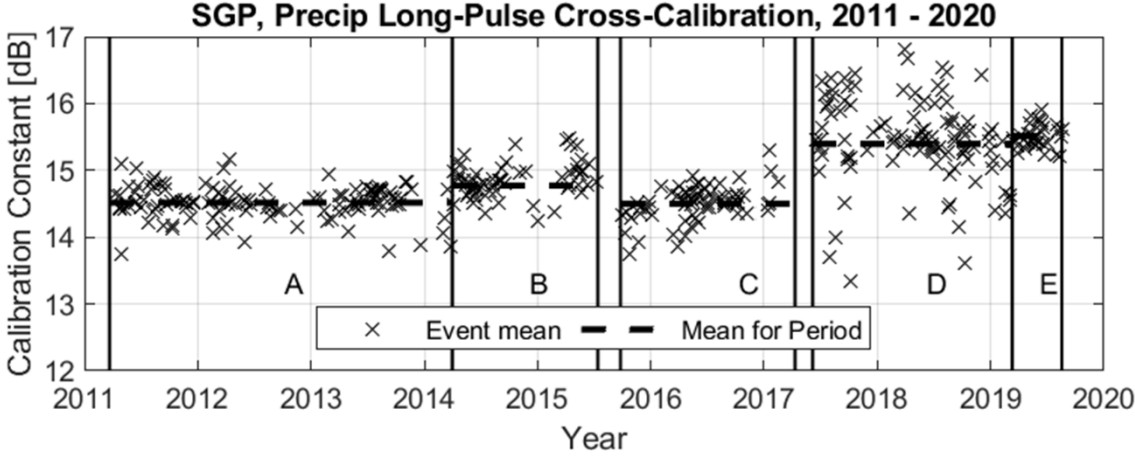

**Figure 13. Precipitation long-pulse beam relative calibration constant $C^{PrecipLong}$ [dB] from 22-March-2011 through 18-August-2019 estimated using individual rain events (black crosses). Thick dashed lines are mean relative calibration constants (listed in Table 4) for stable hardware intervals labelled A through E as described in Table 2.**

**Table 4. RWP relative calibration constants [dB] (standard deviation) relative to precipitation short-pulse mode**

| Period | Start | End | Precipitation Long-Pulse | Wind Mode BeamV | BeamA | BeamB |
|--------|-------|-----|--------------|-------|-------|-------|
| A | 22-March-2011 | 31-March-2014 | 14.5 (0.3) | | | |
| B | 1-April-2014 | 14-July-2015 | 14.8 (0.3) | 9.3 (0.7) | 12.9 (3.1) | 19.9 (2.3) |
| C | 25-Sept-2015 | 10-April-2017 | 14.5 (0.3) | 8.7 (0.7) | 7.8 (1.1) | 6.5 (1.5) |
| D | 6-June-2017 | 10-March-2019 | 15.4 (0.7) | 9.0 (0.9) | 8.1 (2.6) | 2.1 (2.9) |
| E | 11-March-2019 | 18-August-2019 | 15.5 (0.2) | | | |

### 4.2.2 Changes in Wind Mode Relative Calibration Constants

Similar to the conditions applied when estimating the precipitation long-pulse beam relative calibration constants, the wind mode beams were estimated for every day with at least 1000 range gate samples between 500 and 2100 m range and with precipitation short-pulse reflectivity factor greater than 30 dBZ. The wind mode has a shorter pulse length than the precipitation long-pulse beam, which enables valid wind observations down to 500 m. Figure 14 shows the daily relative calibration constants for the three wind beams (black crosses) with thick dashed lines representing the mean relative calibration constant for each hardware interval. The vertical beam relative calibration constant is fairly stable over the 2014 to 2019 observation period, with values listed in Table 4. There is more event-to-event variability in the oblique beam relative calibration constants compared to the vertical beam because there is more horizontal distance between the vertical pointing reference beam and the

oblique beams. A 14° off-vertical pointing angle causes approximately 250 m horizontal distance between the vertical beam and oblique beam at 1 km height. Aside from the larger event-to-event variability, the oblique beam mean relative calibration constants change for each radar hardware configuration. This is probably due to changes in the antenna phase shift module that controls the antenna beam pattern and pointing direction. Table 4 lists the mean oblique beam relative calibration constants for each hardware configuration.

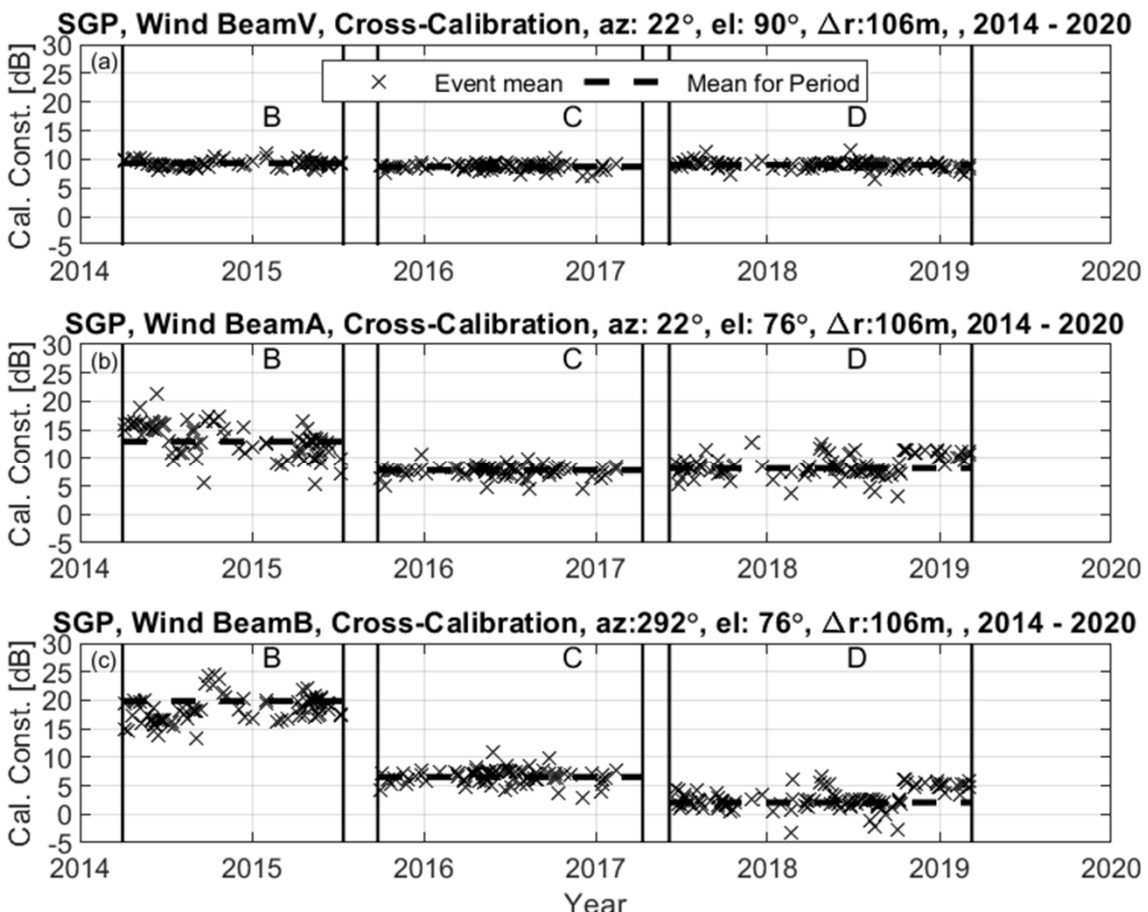

Figure 14. Relative calibration constants for wind mode for every rain event from March 2014 through February 2019. (a) Vertical beam (beamV, az: 22°, el: 90°), (b) oblique beam (beamA, az: 22°, el: 76°), and (c) oblique beam (beamB, az: 292°, el: 76°). Thick dashed lines are mean relative calibration constants (listed in Table 4) for stable hardware intervals labelled B, C, and D as described in Table 2.

## 5 Conclusions

This work describes a procedure to calibrate a UHF band radar wind profiler (RWP) reflectivity factor to surface disdrometer observations. The revised procedure builds on the method described in Tridon et al. (2013) by correcting the recorded Doppler velocity power spectra due to Nyquist velocity aliasing and coherent integration bias effects before recalculating the spectrum

moments. The revised method also calibrates the oblique pointing RWP beams that are used to measure horizontal wind motions.

This cross-calibration procedure uses precipitation measurements from one instrument (i.e., surface disdrometer) as the reference dataset and then calibrates another instrument (i.e., the RWP) using measurements from the same precipitation event. This method cannot identify any biases in measurements from either instrument and the difference in measurements also includes instrument measurement uncertainties. To address biases, the calibration procedure is structured so that a single calibration constant establishes the disdrometer-to-radar calibration. Then, if future comparisons with another instrument

determine the disdrometer-to-radar calibration is biased, a simple offset can be added to the radar reflectivity factor.

       Regarding measurement uncertainties, the standard deviation of the reflectivity factor difference (i.e., $sd[Z^{PrecipShort} - Z^{Disdrometer}]$) includes variability due to different measurement technologies and due to spatiotemporal differences between measurements made at the surface and 500 m above the ground. The radar-to-disdrometer reflectivity factor difference standard deviations were similar in magnitude (i.e., approximately 2 dB) to standard deviations from side-

by-side surface disdrometers measuring the same precipitation event (Tapiador et al., 2017; Wang et al., 2021). Thus, the reflectivity factor difference standard deviation is a relative measure indicating the quality of the comparison and is larger than a calibration constant uncertainty.

       The calibration procedure determined an absolute calibration constant for the precipitation short-pulse beam, which was then called the "reference" beam. The relative calibration between this reference beam and all other beams was determined

enabling all beams to be cross-calibrated to the surface disdrometer, including the RWP oblique pointing beams. The horizontal distance between the vertically pointing reference and oblique pointing beams caused an increase in event-to-event variability in the oblique beam relative calibration constant, as the two radar beams were observing different regions of the same precipitation event.

       The precipitation short-pulse calibration constant changed over the 8-year dataset. The calibration constant tended to

increase over time, corresponding to a decrease in radar sensitivity, consistent with hardware degrading over time. Referencing equation (6), degrading hardware will produce smaller $SNR$ for the same radar reflectivity factor, which is compensated with a larger calibration constant. The radar sensitivity increased significantly (i.e., over 10 dB) when degraded hardware was replaced with new hardware. Between early 2013 and mid-2015, the RWP sensitivity decreased by about 15 dB, for a rate of about 7 dB/year, before a hardware failure in mid-2015. Between 2016 and 2019, the RWP radar sensitivity decreased at a rate

of about 3-to-4 dB/year. The approximate 2 dB calibration standard deviation and the slow change in radar sensitivity implies that the calibration constant can be computed using many rain events over a 1- or 3-month interval.

       To promote the calibration of radar wind profilers and other radar systems, the processing codes used in this study are available on a public GitHub repository (Williams, 2023a) and a public Zenodo repository (Williams, 2023b). This code is being incorporated into the ARM RWP processing suite with the intent of ARM RWP spectra being reprocessed using this

calibration procedure. Also, the 8-years of data processed in this study are available on the ARM Archive as a PI product (Williams, 2023c).

**Appendix A**

This appendix describes the processing steps applied to a spectra profile needed to account for Nyquist velocity aliasing (Section 3.1.1), coherent integration bias (Section 3.1.2), and calculating the spectrum moments (Section 3.1.3). As discussed in Section 3.1.1, spectrum power from targets with true radial velocities greater than the Nyquist velocity will appear to be moving in the opposite direction due to velocity aliasing. The Python code provided in public repositories eliminates velocity aliasing by extending the original spectrum from 0 to $V_{Nyquist}$ to the velocity range $-2V_{Nyquist}$ to $-V_{Nyquist}$ (this segment is called "a" in Fig. 2) and copying the segment from $-V_{Nyquist}$ to 0 to the velocity range $V_{Nyquist}$ to $2V_{Nyquist}$ (this segment is called "b" in Fig. 2). One problem created by copying and appending the original spectrum to itself is that the new spectrum now has two peaks with the same maximum magnitude. One peak is in the $\pm V_{Nyquist}$ velocity range and the other peak is in one of the two extended spectrum velocity ranges. To determine which peak to process, the provided Python code utilizes a prior velocity $V_{prior}$ derived from the previous range gate to select one of the two peaks, which ensures continuity between range gates.

Figure A1 shows the flow diagram to process one spectra profile as implemented in the provided Python code. The processing diagram starts in box 1 in the upper left corner of Fig. A1. In box 2, the original spectrum at the lowest range gate is read into memory. The prior velocity $V_{prior}$ is set to zero (box 3), which effectively assumes the spectrum velocity peak is not velocity aliased in this first range gate. The original spectrum is extended to $\pm 2V_{Nyquist}$ in box 4. Box 5 identifies the two peaks in the extended spectrum. Using $V_{prior}$ as the reference, the peak closest to $V_{prior}$ is selected for further processing (box 6). The integration limits $v_{start}$ and $v_{end}$ define the region containing signal power and are needed to estimate the spectrum moments (e.g., see equation 1). Box 7 estimates the integration limits by starting at the spectrum peak and moving down both sides of the peak until the spectrum magnitude drops below the mean noise level $\bar{n}$ (Carter et al., 1995). Box 8 performs the time-domain averaging (TDA) correction, which is only applied to the signal power above $\bar{n}$ between the integration limits. The spectrum moments are calculated in box 9 and the prior velocity $V_{prior}$ is updated in box 10. If the current range gate is not the last range gate in the profile (box 11), then the next range gate original spectrum is read into memory (box 12) and processing continues in box 4. If the current range gate is also the last range gate in the profile (box 11), then box 13 is executed and estimates the adjusted $SNR$ at all range gates using equation (5). Box 14 estimates the radar reflectivity factor at all range gates. The next profile is selected in Box 15 and processing resumes in box 1.

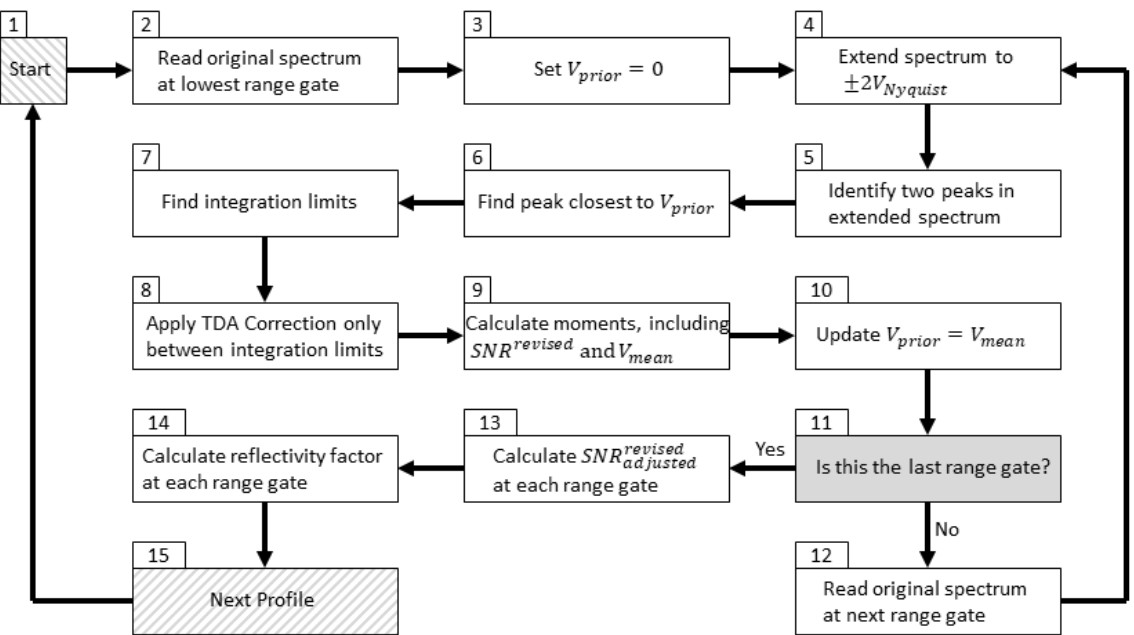

**Figure A1. Flow diagram to process one spectra profile as implemented in the provided Python code.**

*Code availability.* The Python code that processes the raw Doppler velocity power spectra is available on GitHub (https://github.com/ChristopherRWilliams/RWP_Python_Moments). This Python source code is also stored in an open repository with digital object identifier https://doi.org/10.5281/zenodo.7734427.

*Data availability.* All raw observations used in this study are available on-line using the DOE ARM data discovery tool: http://dx.doi.org/10.5439/1025128,          http://dx.doi.org/10.5439/1025129,          http://dx.doi.org/10.5439/1025136, http://dx.doi.org/10.5439/1025137, and http://dx.doi.org/10.5439/1025315.

During the review process, the calibrated RWP moments are available in a DropBox folder (https://www.dropbox.com/sh/d83nvdwg9ouqtz3/AAD-WlKzDQ6ZePfrcSChm3hta?dl=0). A request has been submitted for

DOE ARM to host these data as a PI Product for permanent storage. After DOE has approved this request, the DropBox link will be disabled and this manuscript text will be:

The calibrated RWP moments produced in this study are available on the DOE ARM archive as a PI Product at this link: https://iop.archive.arm.gov/arm-iop/0pi-data/williams/TBD.

*Author contribution.* CRW, PM, and SG conceptualized the study. CRW developed the spectral processing software in the MATLAB language and performed the analysis. JB converted the MATLAB code into Python code. PEJ developed the coherent integration and SNR adjustment methodologies. CRW wrote the manuscript and all authors reviewed and edited the manuscript.

*Competing interests.* The authors declare that they do not have any competing interests.

*Disclaimer.* Authors do not state any disclaimers.

*Acknowledgments.* This research received funding through the US Department of Energy (DOE) Atmospheric System Research (ASR) program under award DE-SC0021345. Additionally, PEJ was supported by NOAA Physical Sciences Laboratory and CIRES through the NOAA cooperative agreement NA17OAR4320101. We recognize and appreciate the work of field technicians deployed year-round to Southern Great Plains and tasked with keeping these instruments running. This research was supported by the Office of Biological and Environmental Research of the US Department of Energy as part of the Atmospheric Radiation Measurement (ARM) Climate Research Facility, and Office of Science scientific user facility.

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
