# Peer review of "Calibrating Radar Wind Profiler Reflectivity Factor using Surface Disdrometer Observations"

_EGUsphere, 2022_

## Author Comment (AC1)

Reply to Reviewer #1:

*Reviewer #2 comments in red italic.*

Author reply in black.

**Blue bold text refers to Track-Change document line numbers.**

Thank you for taking the time to read our manuscript and provide helpful suggestions. Specific comments are entered immediately following the reviewer comments.

*Review of:*

*"Calibrating Radar Wind Profiler Reflectivity Factor using Surface Disdrometer Observations",*

*submitted to the Atmospheric Measurement Techniques (AMT) Journal.*

*This manuscript presents a methodology to calibrate radar wind profiler reflectivity factor using disdrometer data collected during rain events.*

*The authors found that the ageing of the hardware over time greatly impacts the RWP sensibility as it produces smaller values of SNR. The authors proposed compensating for this by varying the calibration constant and several other methods clearly described and evaluated using long-term data sets.*

*It is great to see that the data sets and the code are available (or will be in the near future), thus increasing the reproducibility of the proposed methodology.*

Yes, sometimes, the best way to share information with the community is to provide computer code. It improves long-term availability; the code has been submitted to Zenodo for permanent storage.

*I reckon this manuscript totally fits the scope of the AMT journal, and it is an excellent contribution to the radar community.*

*I have minor comments and suggestions that may help improve the paper.*

*1. The Spectrum adjustment methods and the adjustments due to Nyquist velocity aliasing, coherent integration filtering, and increased noise power will benefit from a diagram. A flowchart describing the critical aspects of the proposed algorithm could help depict the method more clearly.*

Good suggestion; a flowchart would help describe the processing steps. Therefore, Appendix A has been added to the manuscript providing a flowchart and supporting text. The appendix is referenced in Section 3.1. **Please see lines 76, 208-209, 244-248, 313-317, and 600-625, and Fig. A1 in the Track-Change document.**

*2. The sensitivity of the radar plays an important role in the calibration procedure explained in this manuscript. If the RWP cannot collect data during rain events, I reckon this method will not be valid for calibration. I suggest the authors add a discussion regarding this or if some other meteorological target could be used instead.*

Good observation. Yes, the calibration procedure requires observations to be collected during rain or for a mode to be collected during rain and then used as a reference when it is not raining. A paragraph

describing this concern is placed in the Section 2.2 describing the disdrometer data. **Please see lines 164-172.**

---

## Author Comment (AC2)

Reply to Reviewer #2:

*Reviewer #2 comments in red italic.*

Author reply in black.

**Blue bold text refers to Track-Change document line numbers.**

Thank you for taking the time to read our manuscript and provide helpful suggestions. Specific comments are entered immediately following the reviewer comments.

*The manuscript suggests a very interesting methodology to calibrate UHF radar wind profiler data with the help of surface distrometer observations. Getting radar reflectivity values from the hitherto uncalibrated power measurements would clearly increase the usefulness these instruments. This approach furthermore provides an opportunity for a long-term monitoring of the hardware status, which offers very welcome insights as nicely demonstrated by the authors. There are a couple of points that should be addressed in a revised version of the manuscript:*

*As the authors describe, it is known that excessive broadening of Doppler spectra during precipitation leads with neccessity to an incorrect estimation of the noise level by the Hildebrand and Sekhon (1974) method, because this broadening often affects the complete so-called "full-scale" velocity interval (which is bounded by the Nyquist velocity), so that the prerequisites inherent in the noise estimation algorithm are violated. Of course it is possible to address this issue by adjusting the radar settings to allow for a much wider Nyquist limit, in particular by reducing the number of coherent integrations and increasing the length of the time series for the FFT. However, this cannot be done a-posteriori.*

*The authors then assert that the decrease of the signal-to-noise ratio (due to the spectral broadening described above) is due to signal power leakage of the Discrete Fourier Transform (FFT) calculation. Spectral leakage due to the finite extent of time series (in the case of RWP coherently integrated I/Q data) is a well-known fact in Fourier transforms and it is usually controlled through the multiplication of the time series with some kind of window function. However, the authors unfortunately do not provide information on what particular window function was used in the ARM RWP systems. The Vaisala LAP-3000 RWP traditionally use a "von Hann" (or Hanning) window since this operation was quite easy to apply in frequency domain, but other options were possible in later versions of the Vaisala software. The authors should provide this additional detail.*

*However, spectral leakage of the Fourier transform is not the only explanation for the broad Doppler spectra observed with RWP during precipitation: Tests with a similar radar have shown that the broadening of the Doppler spectra is rather independent of the window function used in RWP signal processing, at least as long as a reasonable selection like the Hanning window is made. This observation is in contradiction to the assertion that the broadening is due to spectral processing. Given the dynamic range of the RWP receiver it therefore appears to be more likely that the broadening is caused by signal contributions from the antenna sidelobes. The authors should comment on this alternative explanation, even though it has no consequences for the correction methodology presented.*

Very insightful comments. Thank you for these suggestions. To really understand what processes are causing the increased noise level estimates in the Hildebrand and Sekhon (1974) algorithm (HS algorithm), co-author Paul Johnston performed more simulations producing I and Q voltage time-series

data and then replicating the LAP-XM signal processing chain. The simulations support the reviewer's comments, specifically that the leakage from the FFT is not the cause of the increased noise level estimate. The simulations show that the noise level estimate is biased high when the signal spectrum is so broad that there are not enough noise-only spectral bins for the HS algorithm to get a good statistical representation of the noise. The simulations show that the signal spectrum broadening due to a von Hann time-series window (that is default windowing function in LAP-XM) and FFT processing broadens the signal spectrum by only a couple spectral bins, which is insignificant compared to spectrum broadening due to velocity aliasing and turbulence during intense convective precipitation.

The specific comments about leakage from the FFT processing have been removed from the manuscript. Text has been added to provide more details about how signal spectrum broadening limits the number of noise-only spectral bins in the spectrum and interferes with the HS algorithm making a good noise level estimate. **Please see lines 319-336 in the Track-Change document.**

*A few other minor remarks:*

*Line 16/17: "The third step increases the signal-to-noise ratio (SNR) due to signal power leakage during the Fast Fourier Transform (FFT) calculation" - should be reformulated for more clarity, as the broadening (regardless whether this is due to the antenna sidelobes, or due to spectral leakage) leads to an increase of the noise level and thus a decrease of SNR.*

Agree. There are several processes that increase the estimated noise level more than FFT leakage. The reference to FFT leakage was removed. **Please see lines 18, 188, 329, 334, and 335.**

*Line 34: "At the radar measurement level, radars measure the return signal power as a function of range" - suggest a different wording*

Used different wording. **Please see line 36.**

*Line 37 "every radar subsection" - suggest radar hardware component*

Text changed. **Please see line 39.**

*Line 39 "pole mounted corner reflector calibrations" - this method is impractical for the rather large and fixed phased array antennas of the RWP and does not need to be mentioned here*

Removed text and improved clarity of remaining text. **Please see lines 40-42.**

*Line 57/58: "to account for radio frequency interference (RFI) that sporadically increases noise power estimates" - The effects of RFI remain largely unclear. Therefore the authors should provide more details on the characteristics of this particular RFI contamination. Not every RFI signal increases the noise power.*

Correct, RFI sometimes increases noise power and other times it does not. Reference to RFI interference was removed. **Please see line 61.**

*Line 96 (Table 1 Pertinent RWP operating parameters): For sake of completeness the authors should also provide an short overview of the RWP signal processing algorithms used. Was there any kind of time-domain nonlinear filtering applied ? Was the spectral integration done using Merritts (1995) statistical*

*averaging method? Which moment estimation algorithm was used - single peak picking or multiple peak picking?*

Good idea. A paragraph was added with an overview of the RWP signal processing algorithm. **Please see lines 98-111.**

*Line 243 "The impact of the TDA low-pass filter can be mitigated by applying a correction factor.." The authors could perhaps add a reference to Wilfong et al. (1999) "Optimal Generation of Radar Wind Profiler Spectra" which further discusses the TDA filter characteristics.*

Included the Wilfong et al. (1999) work into this discussion. **Please see lines 282, 292, and 332.**

*Line 277/278: "As the signal power magnitude increases, the FFT leakage causes the spectrum noise power to increase above the noise power produced by other radar noise sources". This statement is rather awkward and should be reformulated, also in view of the remarks made above.*

*Yes, this FFT leakage is not the main cause of the noise power bias. This reference to FFT leakage was removed and the first paragraph in Section 3.2 was rewritten. Please see lines 320-337.*

*Line 312 plus multiple other occasions: "Parson's correlation coefficient" should be corrected to Pearson's correlation coefficient*

Thank you for catching this error. Corrected in 5 places. **Please see lines 375, 377, 383, and 390.**

*Line 645: The given link "https://github.com/ChristopherRWilliams/rwp/Python/spectra" is incorrect. The correct link is obviously https://github.com/ChristopherRWilliams/RWP_Python_moments*

Thank you for seeing this error. The link has been updated. **Please see line 769.**

---

## Author Comment (AC3)

Reply to Reviewer #3:

Reviewer #3 comments in red italic.

Author reply in black.

Blue bold text refers to Track-Change document line numbers.

Thank you for taking the time to read our manuscript and provide helpful suggestions. Specific comments are entered immediately following the reviewer comments.

Williams et al present a methodology for long-term calibration of UHF radar wind profiles using collocated surface disdrometer observations. They apply their method to an 8-year dataset collected at ARMs Southern Great Planes site. The approach is technically sound and the manuscript is well written, hence I recommend the study for publication after very few minor corrections. The open availability of source code and data already during the discussion phase has to be highlighted positively.

**Comments**

L32: It should be stated more clearly, that the assumption of negligible attenuation in rain only holds for the rather long wavelength of the radar wind profiler.

Good suggestion. Text was added to clarify that attenuation needs to be accounted for radars operating at higher frequencies. Please see lines 32-35 in the Track-Change document.

L42f: The list lacks methodologies, where ground-based radars are cross calibrated with collocated ground-based radars, e.g. Hogan et al 2000, Williams 2012, Kneifel et al 2015, and Radenz et al 2018; Also, for the disdrometer comparison, the more recent work of Myagkov 2020 could be cited.

Yes, our omission; cross calibrating against ground-based radars were not included as a method in the manuscript. Good observation. This methodology was added to the manuscript. The recent work of Myagkov (2020) was also added. Please see lines 45-47.

L93: As the scope of the paper is quite technical, it might interest the reader what components where changed following the two hardware failures.

Yes, a reader interested in calibrating radars would also be interested in knowing which components failed. The two failures are included in the text. Please see lines 112-116.

**L199/Fig 3: At first it was confusing, how the algorithm decides, which of the two peaks is the valid one. Only in Sec 3.1.3 L265f, it becomes clear, that the moments of both peaks are calculated and the 'correct' one is only selected later. Please clarify in the text.**

Yes, selecting the 'correct' peak was not described well in the manuscript. An appendix was added to provide more details of the processing steps and to include a flow diagram of the processing steps. The paragraph that contained L199 has been modified. It now describes how the original routine only contained one peak, the extended spectrum has two peaks, and that Appendix A provides details on how a peak is selected. Please see lines 238-251 and the Appendix on lines 600-625.

Also, the word 'correct' has been removed from the manuscript. Please see lines 314, 316, and 317.

L232: The definition of coherence seems quite vague. It should be stated, that the phase difference of the signals has to vary slowly enough.

Text has been modified. Please see lines 270-273.

**Fig 5: It seems that the noise oscillates with season before 2017, but ceases to do so after the hardware change. Have you looked into that issue and is it related to hardware temperature stability?**

Good observation. We had not noticed that the seasonal variability (due to temperature) went away with the new hardware. We looked into the Data Quality Reports and learned that the air condition system for the equipment shelter was updated in the summer of 2016. Yet, the noise contained to have a seasonal variability in 2016. This indicates that the updated transmitter is more stable than the older unit. Text was added to provide this insight. Please see lines 344-347.

**L311: Is -1min time lag typical? Could you identify any dependence on horizontal wind?**

Good point. Yes, there is a dependence on horizontal wind speed and direction. The disdrometer was located about 100 m from the radar and with a 1-minute resolution. There were times when the surface disdrometer data led the radar data collected at 500 m altitude. This could only happen if the horizontal wind was advecting the rain.

A sentence was added to the Introduction and Section 3.3 clarifying that the wind speed and direction dependence affect the lag between the two time-series data. Please see lines 65-68 and 373-375.

*Fig 8: It would help to understand the case study, if the 2DVD derived drop size distribution and rain rate would be shown in the figure as well.*

Another panel was added to Fig.8 showing disdrometer rain rate and mass-weighted mean diameter (Dm) for this event. Please see new Fig. 8.

**Also the limits of the colormap should be similar in Fig 8a and Fig 9c.**

Good suggestion. The colorbar was extended in Fig. 9c to match the range in Fig. 8a. Please see new Fig. 9.

**L392: At least the mean relative offset and the standard deviation should also be given for the wind mode beams. One would suspect, that Beam V shows the least offset, as it is pointed vertically as well? Do the offsets change over time?**

The manuscript is not very clear. The phrase "results not shown due to space limitations" refers to the event scatter plots and event time-height cross-sections shown for the precipitation long-pulse beam in Figs. 10 and 11. The wind mode beam relative offsets, standard deviations, and variations over time are discussed in Section 4, listed in Table 4, and shown in Fig. 14. To clarify the manuscript, the phrase "results not shown…" was removed and a new paragraph was added. Please see lines 462-465.

**L418: From Fig 12 the impression arises, that the calibration was stable until early 2013, then changing rapidly until 2014, afterwards being more stable until mid-2015. This would give a change of about 13dB/year, but other years being more stable.**

The text was changed to highlight the calibration stability in the first few years followed by a rapid decrease in sensitivity. Please see lines 490-501.

Code availability: For long-term availability of the source code, please consider also submitting it to a permanent storage, such as zenodo.

Good idea. Source code added to Zenodo. Please see lines 635-636.

Technical issues

L103: "Pulse duration () [ns]" unnecessary set of brackets

Fixed. Please see line 127.

L533: "[...] on the DOE ARM archive as a PI Product [...]"

Fixed. Please see line 645.

References

*Myagkov, A., Kneifel, S., and Rose, T.: Evaluation of the reflectivity calibration of W-band radars based on observations in rain, Atmos. Tech., 13, 5799–5825, https://doi.org/10.5194/amt-13-5799-2020, 2020.*

*Williams, C. R.: Vertical Air Motion Retrieved from Dual-Frequency Profiler Observations, J. Atmos. Ocean. Tech., 29, 1471–1480, https://doi.org/10.1175/JTECH-D-11-00176.1, 2012*

Radenz, M., Bühl, J., Lehmann, V., Görsdorf, U., and Leinweber, R.: Combining cloud radar and radar wind profiler for a value added estimate of vertical air motion and particle terminal velocity within clouds, Atmos. Tech., 11, 5925–5940, https://doi.org/10.5194/amt-11-5925-2018, 2018.

Hogan, R. J., Illingworth, A. J., and Sauvageot, H.: Measuring Crystal Size in Cirrus Using 35- and 94-GHz Radars, J. Atmos. Ocean. Tech., 17, 27–37, https://doi.org/10.1175/1520-0426(2000)017<0027:MCSICU>2.0.CO;2, 2000

Kneifel, S., von Lerber, A., Tiira, J., Moisseev, D., Kollias, P., and Leinonen, J.: Observed Relations between Snowfall Microphysics and Triple-Frequency Radar Measurements, J. Geophys. Res.-Atmos., 120, 6034– 6055, https://doi.org/10.1002/2015JD023156, 2015